behaviour/evolution

sexual communication, fitness, trade-offs, sex pheromone, Lepidoptera

**Author for correspondence:**
Thomas Blankers
e-mail: thomasblankers@gmail.com

†These authors contributed equally to this study.

# Sex pheromone signal and stability covary with fitness

Thomas Blankers[1,†], Rik Lievers[1,†], Camila Plata[1], Michiel van Wijk[1], Dennis van Veldhuizen[1] and Astrid T. Groot[1,2]

[1]Institute for Biodiversity and Ecosystem Dynamics, University of Amsterdam, Science Park 904, Amsterdam, The Netherlands
[2]Max Planck Institute for Chemical Ecology, Hans-Knöll-Strasse 8, Jena, Germany

TB, 0000-0002-1893-8537; ATG, 0000-0001-9595-0161

If sexual signals are costly, covariance between signal expression and fitness is expected. Signal–fitness covariance is important, because it can contribute to the maintenance of genetic variation in signals that are under natural or sexual selection. Chemical signals, such as female sex pheromones in moths, have traditionally been assumed to be species-recognition signals, but their relationship with fitness is unclear. Here, we test whether chemical, conspecific mate finding signals covary with fitness in the moth *Heliothis subflexa*. Additionally, as moth signals are synthesized de novo every night, the maintenance of the signal can be costly. Therefore, we also hypothesized that fitness covaries with signal stability (i.e. lack of temporal intra-individual variation). We measured among- and within-individual variation in pheromone characteristics as well as fecundity, fertility and lifespan in two independent groups that differed in the time in between two pheromone samples. In both groups, we found fitness to be correlated with pheromone amount, composition and stability, supporting both our hypotheses. This study is, to our knowledge, the first to report a correlation between fitness and sex pheromone composition in moths, supporting evidence of condition-dependence and highlighting how signal–fitness covariance may contribute to heritable variation in chemical signals both among and within individuals.

## 1. Introduction

Many sexually reproducing organisms discriminate among potential mates. By selecting a mate, choosing individuals may receive direct benefits, e.g. protection or nutrients, and indirect benefits, e.g. by receiving 'good' genes which result in more viable or sexy offspring [1–5]. Mate choice commonly occurs through sexual signals, and variation in reproductive success across individuals producing different sexual signals is the basis of sexual selection [3,6,7].

In general, signals under sexual selection are subject to directional selection, as those individuals are chosen that confer the highest direct or indirect benefits. Additionally, in many organisms, sexual signals are used to localize potentially suitable (conspecific) mates, often referred to as species recognition [6,8]. These so-called species-recognition signals are often under stabilizing selection, because variation in these signals renders them less reliable. Both directional and stabilizing selection are expected to erode genetic variation [9,10]. However, sexual signals are shown to have high levels of genetic variance [11,12] and many observations indicate that sexual signals evolve rapidly, diverge early on during speciation and are important barriers to gene flow among closely related species [6,13–16]. To understand how sexual signals evolve, it is important to understand how (genetic) variation in these signals is maintained.

If signals are costly to produce or maintain, their expression and their composition are expected to be correlated with fitness [5,17]. Negative correlations between signal and fitness indicate that signal investment trades off with fitness (one-trait trade-offs *sensu* [18]). Positive correlations between signal expression and fitness are expected when only high-quality senders are able to bear the cost of the signal (two-trait trade-offs *sensu* [18]) and indicate that the signal is condition-dependent [19]. Covariation between sexual signal variation and fitness can maintain genetic variation in sexual signals, even in the face of selection [20–23]. This is because fitness is the result of the combined effect of many different traits and thus controlled by many different loci [24].

Signal–fitness covariance has been studied mostly in species with acoustic and visual signals, while chemical signals have received much less attention [25,26]. This may be because chemical signals, such a sex pheromones, are generally assumed to be biosynthetically cheap [25,27,28] and have traditionally been assumed to be independent of signaller quality [25,29]. Specifically, moth sex pheromones have typically been considered as species-recognition signals [30,31] and empirical evidence suggests moth pheromone signal composition is under stabilizing selection [32–36]. However, sexual signals probably do not function solely as species recognition or mate choice signals, but rather range along a continuum [8,21,37]. This idea is supported by empirical studies that showed a relationship between nutrition and pheromone amount [38], body size and sex pheromone amount [39], and body size and sex pheromone composition [40] in moths, and between nutritional state, age, and parasite load and pheromone composition in beetles [41,42]. However, even though moth sex pheromones have been studied extensively in the past 40–50 years and the sex pheromone of greater than 2000 moth species has been studied [43], very little is known about the relationship between signal variation and fitness or about variation within individuals. Moreover, we lack empirical insight into the relationship between sexual signal variation and fitness for chemical mate attraction in insects generally.

Here, we tested the hypothesis that even for stereotypical species-recognition signals, there is covariation between sexual signal composition and fitness. Because many sexual signals need to be maintained throughout an individual's reproductive lifetime, and maintenance probably requires continuous investment, we also hypothesized that the ability of an individual to maintain its signal covaries with fitness as well. We tested these hypotheses for the female sex pheromone in the noctuid moth *Heliothis subflexa*. Specifically, we examined how sex pheromone signalling activity (i.e. the time spent sending the pheromone signal, calling activity from hereon), and pheromone amount and composition changed over time.

In moths, older (virgin) females tend to have reduced mating activity and reduced mating success [44–50], and virgin female moths generally keep investing in signalling [51]. Thus, prolonged virginity is expected to trigger physiological responses that modulate resource allocation between somatic maintenance and reproduction. We assessed these trade-offs by comparing a biologically realistic delay to first mating (3 days) with an extreme case of prolonging female virginity (8 days), here referred to as 'early' and 'late' maters, respectively. We then asked whether (i) the composition and amount of the pheromone signal in early and later maters covaried with fitness, and (ii) the stability (within-individual variation) of the pheromone during prolonged virginity was correlated with fitness. We addressed these questions separately in the early and in the late maters to explore the robustness of our findings to (i) variation in age and (ii) the time span over which intra-individual variation was measured. We expected that high-fitness females produced more pheromone compared to low-fitness females, and that maintaining high pheromone amounts and stable pheromone composition trades off with fitness.

## 2. Methods

### 2.1. Insects

The laboratory population of *H. subflexa* originated from North Carolina State University (NCSU) and has been reared at the University of Amsterdam since 2011, with occasional exchange between NCSU,

Amsterdam and the Max Planck Institute for Chemical Ecology in Jena to maintain genetic diversity. The rearing was kept at 25°C and 60% relative humidity with 14 L : 10 D light–dark cycle. Larvae were reared on a wheat germ/soy flour-based diet (BioServ Inc., Newark, DE, USA) in 37 ml individual cups. Pupae were separated by sex and checked for emergence every 60 min from 2 h before until 7 h after the onset of scotophase. Only females that emerged within this timeframe were included in the experiments. After emergence, females were transferred to clear 475 ml observation cups covered with fine mesh gauze. Males were kept in the 37 ml individual cups. Adults were provided with regularly replaced cotton soaked in 10% sugar water.

## 2.2. Phenotyping

The pheromone of each female was sampled at two time points. The first sample was taken in the first night after emergence. The second sample was taken in the third or eighth night after emergence for the 'early' and 'late' treatment, respectively. Early maters were females that were kept virgin for 3 days and mated on the fourth day, while late maters were females that were kept virgin for 8 days and mated on the ninth day. To avoid possible variation owing to differences between sampling time during the night [52–54], care was taken to take the first and second pheromone sample at the same time during the night (no more than 10 min difference) and the sex pheromone was always sampled within a narrow time interval during peak calling times (third to sixth hour of scotophase [52].

Pheromone was collected from the gland surface from females using optical fibres coated with a 100 µm polydimethylsiloxane (Polymicro Technologies Inc., Phoenix, AZ, USA), as described in detail in [55], where the authors showed strong correspondence between pheromone measurements following the non-invasive method used here and following traditional (lethal) gland extractions. Pheromone glands were extruded and fixed by gently squeezing the abdomen and gently rubbed over a period of 2 min. Fibres were then submerged for 1–2 h and rinsed in 50 µl hexane with 200 ng pentadecane as internal standard and discarded. Pheromone extracts were analysed by injecting the concentrated samples into a splitless inlet of a 7890A gas chromatograph (GC) (Agilent Technologies, Santa Clara, CA, USA) and integrating the areas under the pheromone peaks, using Agilent ChemStation (v. B.04.03).

Throughout this study, we focused on the sex pheromone components that have been shown to be important for male attraction—(Z)-11-hexadecenal (Z11-16:Ald), (Z)-11-hexadecenyl acetate (Z11-16: OAc), (Z)-11-hexadecenol (Z11-16:OH), and (Z)-7- and (Z)-9-hexadecenals (Z7-16:Ald + Z9-16:Ald)—as well as the total amount of pheromone. Z7-16:Ald and Z9-16:Ald were difficult to separate by GC and were, therefore, integrated as one peak (referred to as Z7/Z9-16:Ald), as has been done in previous studies [56]. Absolute amounts (in ng) and percentages (of the total amount) of each compound were calculated relative to a 200 ng pentadecane internal standard. Samples containing less than 10 ng could not be reliably integrated and were excluded from subsequent analyses. We globally standardized the absolute amount of each of the four pheromone components by subtracting the global average amount across all females and then dividing the mean-centred value by the variance. We then performed principal component analysis on the standardized pheromone measurements, retaining four principal components (PCs). Differences in PC scores between the first and second pheromone samples were tested using a paired Student's *t*-test, using the statistical package R [57].

## 2.3. Relationship between fitness and female calling activity

To determine whether calling activity of virgin females increased or decreased over their lifetime or whether it peaked at some point in their reproductive life, we randomly assigned 40 and 51 females to the 'early mating' and 'late mating' treatments, respectively, and observed the calling activity of each female every night from eclosion until her death. For each female, we measured pupal mass, calling behaviour, fecundity, fertility and lifespan. Pupae with a visible wing pattern were weighed before the start of scotophase (the 10 h dark phase during the day–night cycle). Extremely small pupae (less than 0.1 g) were discarded. The time spent calling before and after mating was recorded daily every 30 min between 1 and 8 h after the onset of scotophase. To measure fecundity and fertility, each female was mated in a 475 ml observation cup with a randomly chosen, virgin 2- to 3-day-old male 1 day after the second pheromone sample was taken. For all mated females, we scored the number of eggs, the number of hatching larvae and the lifespan. To stimulate oviposition, freshly cut *Physalis peruviana* berries were supplied daily onto the gauze. Before the onset of a scotophase, eggs laid in the previous scotophase were collected by transferring the female to a new observation cup.

Eggs were counted before emergence, and artificial diet was provided for emerging larvae. Emerged larvae were counted daily. After female death, lifespan was recorded, and mating success was confirmed by checking for the presence of a spermatophore. The percentage of calling females, the per-individual onset of calling and the per-individual duration of calling were visually examined before and after mating. We tested whether early maters differed from late maters in fecundity (overall and per day), fertility (overall and per day) and lifespan using two-sample $t$-tests.

## 2.4. Relationship between fitness and *inter*-individual variation

To test whether fecundity, fertility and lifespan variation could be predicted by variation in the pheromone, we fitted generalized linear models with quasi-Poisson error and used a stepwise model selection approach to identify significant predictors and interactions. In the base model, the response was either fecundity, fertility or lifespan and the predictors were each of the four PCs as well as the pupal mass of the female, the pupal mass of her mate and the time spent calling by the female before mating. Only PC scores from pheromone measurements taken on timepoint 2 were used, as these were the samples taken immediately before the matings were set up that were used to measure the individual's fitness. Model selection was done by first adding interactions among the independent variables and then purging independent variables that did not significantly explain variation in the response. Adding and removing variables and interactions was done using analysis of deviance, where significance of an increase/decrease in residual deviance after removing/adding a variable or interaction was assessed using a $\chi^2$ test. Improvement of model fit was considered significant for $p < 0.05$. For the final model, the pseudo-$R^2$ was calculated as the ratio between the null deviance and 1 minus the residual deviance, similar to calculating the coefficient of determination in linear regression. The pseudo-$R^2$ gives the proportion of deviance explained, which is informative about model fit and about the explanatory value of the independent variables [58].

## 2.5. Relationship between fitness and *intra*-individual variation

To test whether fecundity, fertility and lifespan variation could be predicted by the degree to which females kept their pheromone stable, we calculated intra-individual variation in pheromone by taking the absolute difference between PC scores for pheromone measurements taken after 24 h and PC scores for pheromone measurements taken 3 or 8 days post-eclosion, for early and late maters, respectively. We fitted the same generalized linear models (with the same covariates) and employed the same model selection procedure as described above, except that the predictor variables were now the absolute change in PC scores between timepoint 1 and timepoint 2.

## 2.6. Heritability of stability

To determine the heritability of pheromone composition (PC scores) and stability (difference between PC scores at timepoint 1 and 2), we selected 20 females that covered the range of intra-individual variation observed across all females. Subsequently, we sampled 5–20 daughters per female ($n = 240$) at both timepoints, i.e. 3 or 8 days post-eclosion for early and late maters, respectively. We used Bayesian Monte Carlo Markov chain (MCMC) models combined with pedigree information of the two generations of females and their mates. We ran independent models for each PC (inter-individual variation) or for the change in PC scores between timepoints 1 and 2 (intra-individual variation). The models were implemented with the R package 'MCMCglmm' [59] using inverse γ priors. All MCMC models were run for 1 000 000 iterations, the initial 100 000 samples were discarded (burn-in period) and chains were sampled every 100 iterations (thinning) to reduce autocorrelation. The narrow-sense heritability ($h^2$) was estimated as the ratio of additive genetic variance ('animal' effect) over the sum of all variance compounds ('animal' plus 'unit' effects).

# 3. Results

## 3.1. Relationship between fitness and female calling activity

When observing female calling activity, we found that calling effort of all females and duration of individual calling activity initially increased and then decreased after the second or third night (figure 1$a,b$). Upon mating, calling activity was almost entirely suspended, after which it slowly

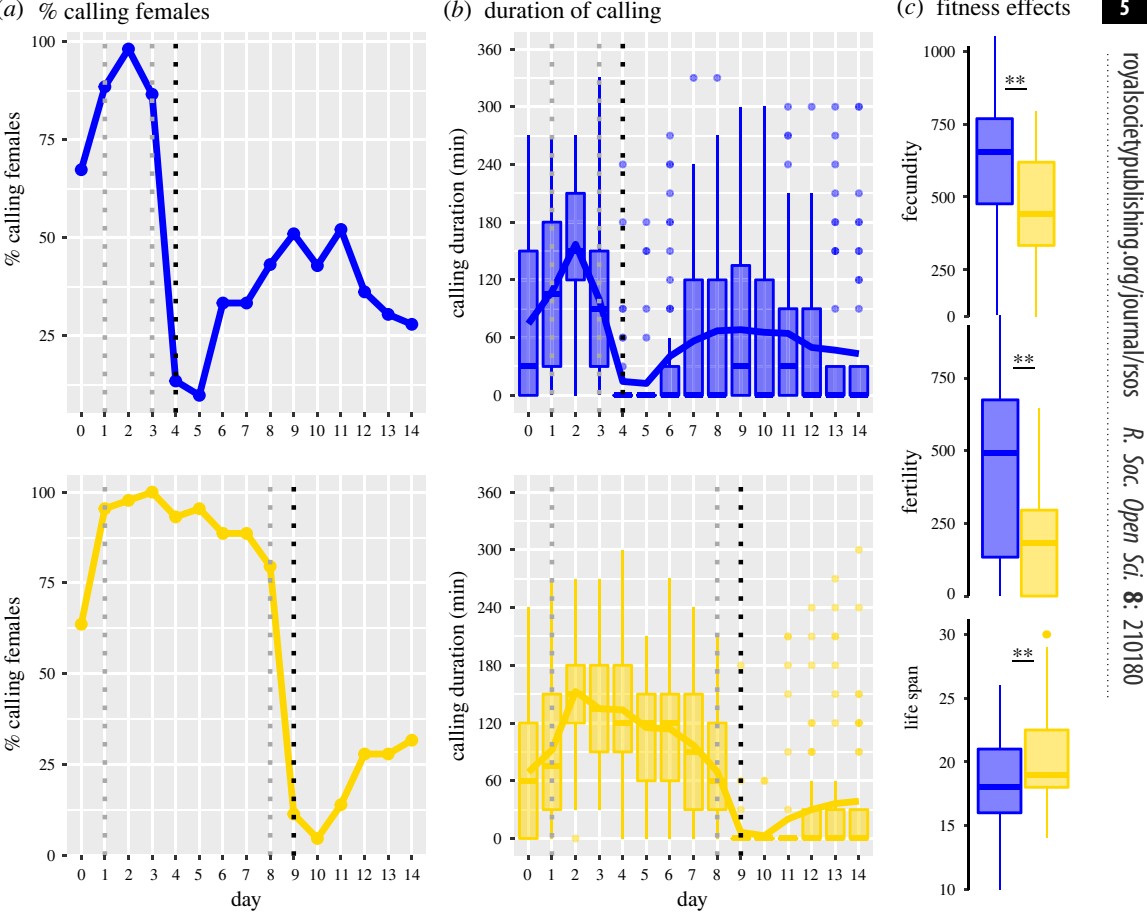

**Figure 1.** Calling activity and fitness. For both groups, early (blue) and late (yellow) maters, the collective calling effort per night and the distribution of calling duration across females is shown for each night. (*a*) Per cent of females calling on any given night, (*b*) calling duration. Boxes and whiskers show the interquartile ranges and the median, and lines connect the mean values for each night. Individual dots show observations greater than 1.5 times the interquartile range. The dashed, vertical grey and black lines show the days at which pheromone samples were collected and females were mated, respectively. (*c*) Fitness variation within and between early and late maters. $^{**}p < 0.01$.

recovered. There was no apparent relationship between the time at which females started calling and female age or mating status (electronic supplementary material, figure S1).

Late maters invested in pheromone calling for an additional 5 days relative to early maters (figure 1). Both fecundity and fertility were significantly lower in late maters relative to early maters (figure 1*c*). Reduced fecundity was probably owing to the 'lost' 5 days rather than owing to allocation of time and energy, because fecundity per day was not significantly different between early and late maters (46.50 versus 40.96 eggs d$^{-1}$; $t = 1.39$; $p = 0.1669$). Fertility per day was lower in late versus early maters (19.00 versus 32.84 larvae d$^{-1}$; $t = 3.102$; $p = 0.0026$), indicating that females who mated later in life laid eggs with lower hatching success compared to early maters. We further observed that late-mated females lived on average 2 days longer than early-mated females (figure 1*c*).

## 3.2. Variation in sex pheromone signal strength and composition

In quantifying the sex pheromone variation using PC scores for the standardized absolute amounts of four components that are important for male response in *H. subflexa*, we found that the first PC accounted for 60.5% of the total variation among individuals. Because this axis was almost perfectly correlated with the total amount of pheromone (Pearson's $r = -0.99$; $p < 0.0001$), PC1 thus described variation in the total amount of pheromone. The remaining three PCs described different dimensions of pheromone composition space: higher scores on PC2 (19.8% var. expl.) indicated decreasing amounts of Z11-16:OAc relative to all other compounds, higher scores on PC3 (13.2% var. expl.) indicated increasing amounts of Z11-16:OH relative to Z7/Z9-16:Ald, and higher scores on PC4

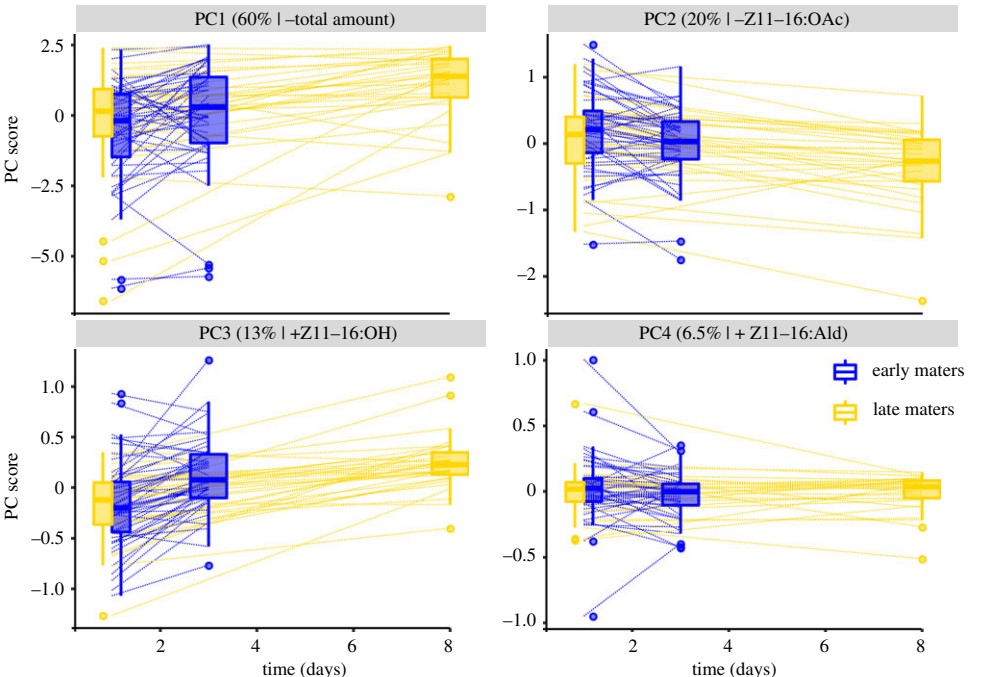

**Figure 2.** Pheromone variation within and between females. Box-and-whisker plots show the distribution of PC scores at day 1 and at day 3 or 8 for early (blue) and late (yellow) maters, respectively. Dashed lines connect samples taken from the same individual. For each PC, the amount variance explained and the main pheromone component that loads on the PC, as well as the sign of the loading (positive, +, or negative, −), is indicated above each panel.

(6.54% var. expl.) indicated increasing amounts of the major component, Z11-16:Ald, relative to all other compounds (electronic supplementary material, table S1 and figure S2). Neither of these three 'composition' PCs were correlated with the total amount of pheromone (Pearson's $r$ ranged from −0.05 to 0.08; $p$ ranged from 0.3337 to 0.4987) and a significant proportion of the variation in all PC scores was additive genetic variance, except PC1 in the late maters (posterior mode of $h^2$ ranged from 0.17 to 0.58; for PC1 in late maters $h^2 < 0.01$; electronic supplementary material, figure S3). There was thus no need to calculate log-contrasts for the components to break the unit-sum constraints that are problematic in the analysis of relative amounts [60], thereby avoiding inadvertent effects resulting from the choice of the divisor. For completeness, relative and absolute amounts for all pheromone measurements as well as life-history data are given in the electronic supplementary material, table S2.

With the exception of PC4, all PCs were significantly different between timepoint 1 and timepoint 2, both for early and late maters (figure 2). This shows that there is intra-individual variation in both amount and composition of the pheromone signal. Over time, pheromone amount decreased, while relative amounts of Z11:16:OAc and Z11-16:OH increased (figure 2). However, there was substantial variation around the mean direction and magnitude of change, with some individuals showing much more or much less change in the amount and composition of the pheromone over time (figure 2). This shows that there is variation in the magnitude of intra-individual variation. Estimates of the narrow-sense heritability of intra-individual variation in the PCs were between 0.1 and 0.2, indicating that a significant proportion of the variation is additive genetic variation (electronic supplementary material, figure S3).

## 3.3. Relationship between fitness and *inter*-individual pheromone variation

When we determined correlations between the female pheromone signal and her fecundity, fertility and lifespan, we found that all fitness measurements depended on variation in the pheromone signal. In all models, both in early and late maters, we found at least one PC describing variation in the composition of the pheromone blend to explain variation in fitness (figure 3a). For fecundity (in both early and late maters), we also observed a correlation with PC1. Partial pseudo-$R^2$ values ranged from 3 to 30% for the pheromone components. Time spent calling before mating and the pupal mass of the female also explained fitness variation in most models and the combined effect from the PCs, covariates and their

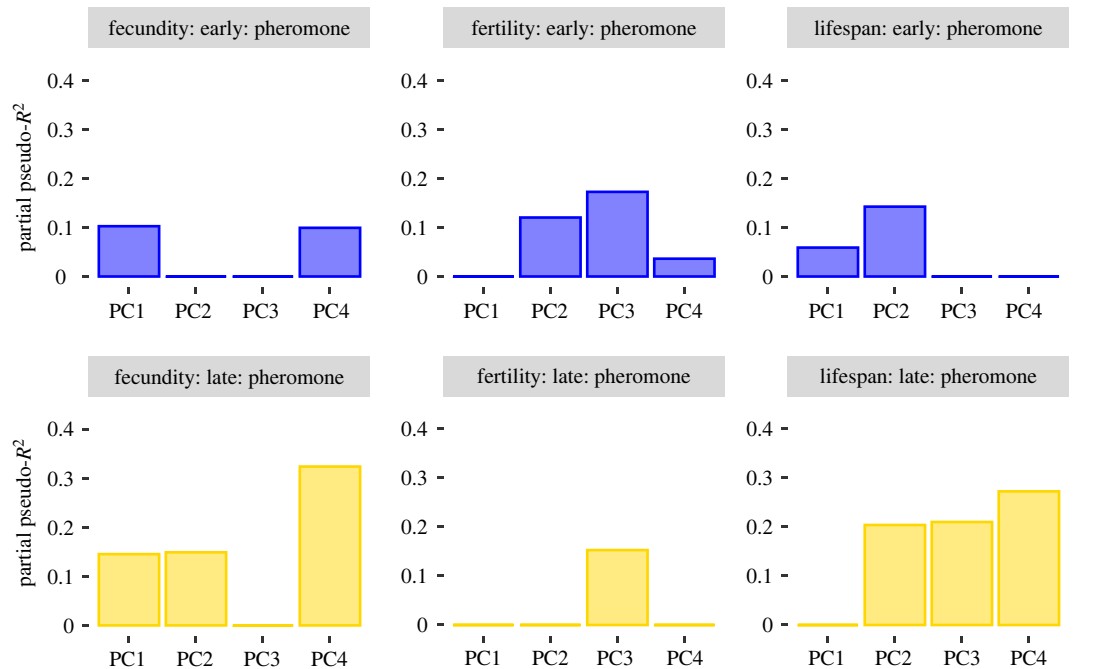

**Figure 3.** Fitness variation explained by pheromone traits. Each panel depicts one model for either early (blue) or late maters (yellow), for fecundity, fertility or lifespan as a response variable, and for pheromone amount and composition as a predictor. For each of the four PCs, columns depict the partial pseudo-$R^2$ of the PC and all interactions in which the PC is involved in the model.

interactions explained between 15 and 59% of the variation in fitness measurements (electronic supplementary material, table S3).

Producing more pheromone (lower scores on PC1) or higher relative amounts of Z11-16:OAc (lower scores on PC2), Z11-16:OH (higher scores on PC3) and Z11-16:Ald (higher scores on PC4) was universally associated with higher fitness (higher fecundity and fertility, longer lifespan) with one exception: higher scores for PC4 were associated with lower fecundity in the early maters (figure 4; electronic supplementary material, figure S4 and table S3). In the case of interactions between a pheromone PC and female pupal mass, we examined the modulating effect of pupal mass on the correlation between fitness and pheromone by drawing separate regression lines for the females with the lowest 33% pupal mass, those with pupal mass in the middle tercile and those in the top 33% of pupal mass, using the R-package 'interactions' [61]. This showed that higher relative amount of Z11–16:OH (higher PC3 score) was associated with higher fertility in the heaviest 33% females, but not in the bottom 67% (figure 4). Also, higher relative amount of Z11-16:OAc (lower PC2 score) was associated with longer lifespan in heavy females, but with lower lifespan in lighter females (figure 4).

## 3.4. Relationship between fitness and *intra*-individual variation

To test the hypothesis that females with stable pheromone signals during their lifetime have higher fitness, we calculated intra-individual variation as the absolute difference between PC scores for pheromone measurements taken 24 h post-eclosion and for pheromone measurements taken 3 or 8 days post-eclosion, for early and late maters, respectively. We found that between 6 and 45% of the variation in fitness measures was predicted by intra-individual variation in either the total amount (PC1), the relative amount of Z11-16:OAc (PC2) or both (figure 5).

In early maters, more stability in the total amount (PC1) was associated with higher fecundity, fertility and longevity, while in late maters, stability in total amount was uncoupled from fitness (figure 6; electronic supplementary material, figure S5, and table S4). For PC2, we similarly found a predominantly positive correlation between stability and fitness (i.e. lower values on the plasticity axis are associated with higher values on the fitness axis). We also observed examples of effects in the opposite direction for both fertility and lifespan (figure 6; electronic supplementary material, figure S5).

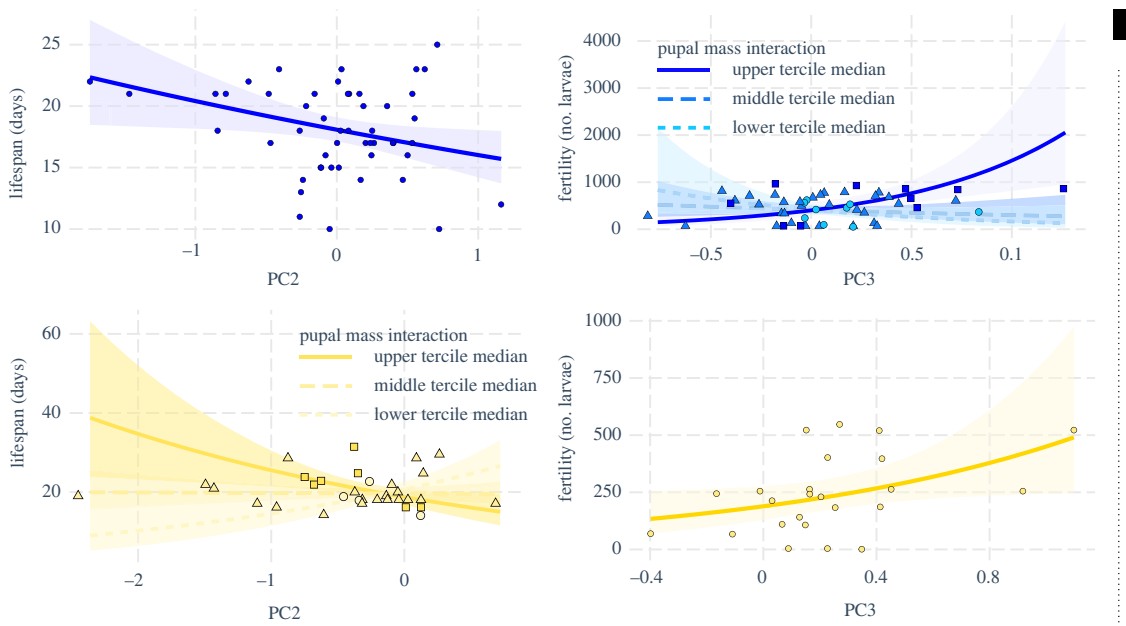

**Figure 4.** Relationship between pheromone variation and fitness, showing the correlation between PC scores and fitness from both early maters (blue) and late maters (yellow). Points show individual coordinates; lines show regression paths, i.e. the effect of the PC (*x*-axis) on fitness (*y*-axis), accounting for the effects from all other predictors in the model. Higher scores on PC2 correspond to lower relative amounts of Z11-16:OAc, while higher scores on PC3 correspond to higher relative amounts of Z11-16:OH. When significant interaction effects between PCs and pupal mass were detected, relationships between PCs and fitness were determined separately for females with pupal masses in the lower tercile (median of the lower tercile = 0.224 g, circles), middle tercile (median = 0.238 g, triangles) and upper tercile (median = 0.275 g, squares).

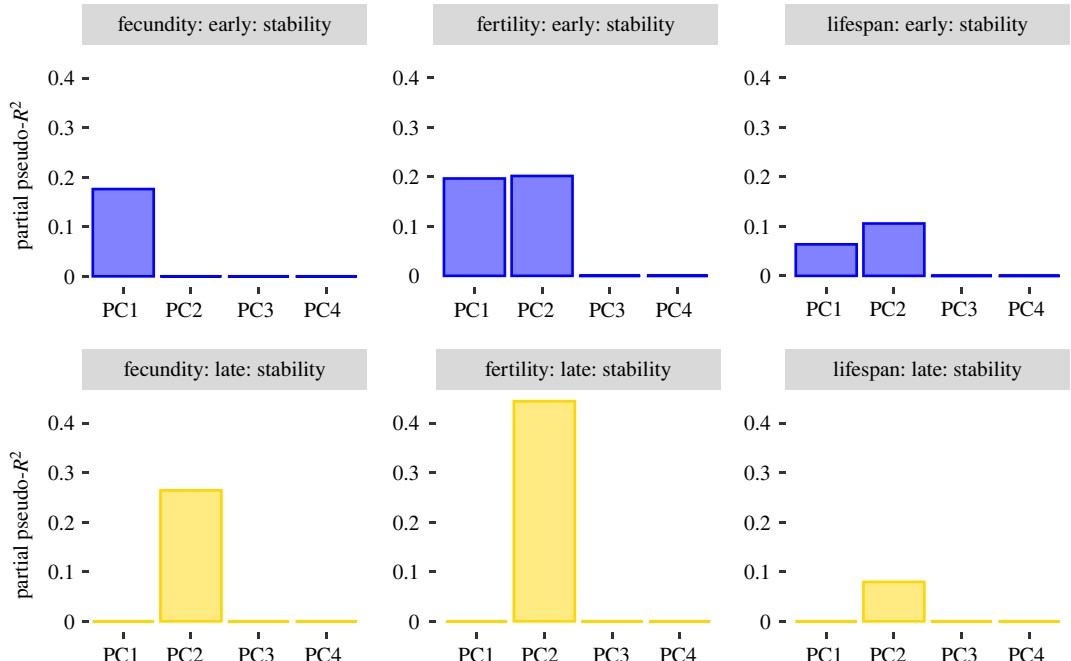

**Figure 5.** Fitness variation explained by intra-individual variation in pheromone amount and composition (here referred to as stability). Each panel depicts one model for either early (blue) or late maters (yellow), for fecundity, fertility or lifespan as a response variable, and for intra-individual variation in pheromone PCs as predictors. Column height corresponds to the partial pseudo-$R^2$ of intra-individual variation in the PC and all interactions in which the PC is involved in the model. PCs absent from the model are shown as partial pseudo-$R^2 = 0$.

**Figure 6.** Relationship between intra-individual pheromone variation and fitness. The magnitude and direction of the correlation between change in PC scores between timepoint 1 and timepoint 2 and fitness in early maters (blue) and late maters (yellow) is shown. When significant interaction effects between PCs and pupal mass were detected, relationships between PCs and fitness were determined separately for females with pupal masses in the lower tercile (median of the lower tercile = 0.224 g, circles), middle tercile (median = 0.238 g, triangles) and upper tercile (median = 0.275 g, squares).

# 4. Discussion

If sex pheromone signals are costly, calling activity, pheromone amount and/or pheromone composition should covary with fitness. We showed that in *H. subflexa*, an extreme delay in mating of 8 days and a continued investment in signalling was associated with lower reproductive output. We also found that while virgin females maintained high signalling activity, their signal changed over time: the total amount decreased and the ratios of components changed. Signal variation was proportional to the delay in mating, but varied considerably across females. Some females showed high intra-individual variation, while others had stable signals over time. The heritability estimates of the intra-individual variation in pheromone composition (roughly between 0.1 and 0.2) indicated that this variation can evolve in response to selection. Lastly, we found that longevity, fecundity and fertility were correlated with pheromone amount, with pheromone composition, and with the stability in both pheromone amount and composition within females. We thus conclude that our results meet expectations for costly pheromone signalling and that the amount, composition and maintenance of the long-distance mate attraction pheromone produced by *H. subflexa* females probably depend on the genetic quality of the female.

## 4.1. Calling activity and fitness

In a synthesis of empirical results on calling activity in female moths, the majority of moth species were found to increase their calling effort over time [51]. This finding is in line with theoretical predictions for a costly signal, for which the production depends on the number of male arrivals: a virgin female should not invest much in signalling if this draws in too many males, but she should increase signalling efforts if no males visit at all [51]. Our results show that for *H. subflexa*, calling activity initially indeed increased, but then decreased (figure 1). Females that were mated later spent more nights calling and had reduced reproductive success, but lived slightly longer relative to females that were mated earlier in life. These findings are in line with general patterns across moths [62] and suggest that the additional days spent calling in the slightly longer-lived late maters reduce the energy and time available to lay eggs, implying costs to calling. However, we acknowledge that there may be confounding effects from reproductive senescence on the fitness differences between early and late maters.

## 4.2. Pheromone variation and fitness

We hypothesized that these costs would manifest in signal–fitness correlations in virgin moths. We found evidence for covariation between fitness and sex pheromone amount, composition and stability, regardless of the fitness measure (reproductive output or lifespan). In addition, we found similar patterns in two independent groups of females that differed in their age at the time of sampling and in their remaining lifespan to lay eggs. Even though in nature females are unlikely to stay virgin for 8 days (late maters), our finding that her signal still correlates with fitness shows that the patterns that we found are independent of female age. Our findings thus strongly support a scenario for signal–fitness covariance in moths and provide context to earlier findings that suggested condition-dependence of moth sex pheromone amount [38,39] and composition [40].

On the one hand, the consistent positive correlations between fitness and pheromone amount as well as between fitness and the relative abundance of pheromone components important to male mate attraction are surprising from a physiological perspective. It may be unlikely that pheromone production presents significant metabolic costs to signalling females, because the nutrients used for pheromone production are negligible compared to available resources [28] and storage of pheromone titres does not seem to be constrained by the level of synthesis, but rather by the level of breakdown (catabolism) [63]. However, sugar feeding does increase pheromone titres [64,65] and starved females produce low titres and are less attractive [65]. In addition, there may be indirect costs, for example, if enzymes that convert sex pheromone component precursors into their final products (in the appropriate ratios of species-specific blends) are also used for other critical physiological processes.

On the other hand, the fitness costs to sex pheromone composition inferred here can explain field observations of sex pheromone variation among wild populations of *H. subflexa*. The acetate ester Z11-16:OAc slightly improves the attractiveness of the *H. subflexa* blend, while strongly deterring the closely related tobacco budworm, *Heliothis virescens* [66,67]. In regions where *H. virescens* co-occurs with *H. subflexa*, the relative amount of Z11-16:OAc of field-caught females is significantly higher compared to regions where *H. virescens* is absent [68]. Apparently, Z11-16:OAc is produced at higher rates when communication interference is likely to happen. As we found that the PC2 axis, associated with variation in Z11-16:OAc, explained 10–20% of the variation in fecundity, fertility and lifespan (figure 3), such that females with higher fitness had higher relative amounts of Z11-16:OAc, selection probably favours females that produce lower relative amounts of acetate esters when there is no risk of heterospecific mate attraction.

There are limited data available for male response to intra-specific levels of variation in moth sex pheromones and these data often come from different methods, including sticky traps, field traps and wind tunnel assays. It is, therefore, difficult to judge the relevance of the levels of observed variation to male response. The two axes that described most of the variation in pheromone composition (and that correlated with fitness), PC2 and PC3, are driven by variation in Z11-16:OAc, Z11-16:OH and Z9-16:Ald, which all have been shown to be important for male attraction [66,67]. The relative amount of Z11-16:OAc ranged from 5 to 15%. Previous studies have shown that (i) as little as 1% is enough to completely deter the heterospecific *H. virescens* [69], (ii) adding Z11-16:OAc to a synthetic blend increases the attractivity of *H. subflexa* males [66,70], and (iii) increased levels of acetate esters (from roughly 1–3 to 10–15%) also increased male response [67]. For Z11-16:OH, the lowest and highest scores on PC3 corresponded to a twofold difference, i.e. from approximately 3% to approximately 6%. Previous studies have shown that increasing or reducing the relative amount of Z11-16:OH by a factor of two can reduce male attraction to sticky traps by 10% in the field [71], while in wind tunnel experiments, a change in male attraction was not found [66]. Finally, for Z9-16:Ald, we found a range from approximately 15% to approximately 35%. Previous studies have found a 38% reduction in male precopulatory behaviour when relative amounts of Z9-16:Ald were below 15% [72], while in wind tunnel assays, Z9-16:Ald levels below 10% reduced male attraction by 20% [66]. Hence, the observed levels of variation in the pheromone summarized by PC2 and PC3 fall within a range that is probably relevant for variation in male responses.

We also hypothesized that for costly signals, maintenance is costly too and we thus expected the stability of the signal to covary with fitness. In line with our expectations, we found that individuals with more stable sex pheromone signals over time had higher fitness, although negative correlations were found as well. Several aspects of animal signals may change with age, and the signals of younger individuals have been found to be more attractive to the receivers, for example, in cricket songs and mouse urinary protein pheromones [73,74]. Because in moths, older females are typically less attractive, as measured by mating success [44–50], it is possible that a female benefits from

maintaining a young-female-like blend. In addition to changes in the composition, female moths also produce more pheromone earlier in life. Possible physiological explanations for decreasing pheromone amounts in ageing females are that changes in juvenile hormone may result in the suppression of pheromone production later in life [75,76] or that older females may have a reduced capability to synthesize pheromone components [76]. Lower pheromone amounts can reduce the attractiveness of the blend; however, there is mixed evidence for this hypothesis [51]. Therefore, an evolutionary mechanism that would explain a benefit to costly stability is that stability counters signal decay owing to senescence.

In summary, we find evidence for signal–fitness covariation, which may imply costs to sex pheromone signalling. Our results are in line with earlier findings for condition-dependence of pheromone signals in moths and beetles [40–42] and for signalling activity-fitness trade-offs in moths [39]. Our results go beyond these findings by revealing (i) a relationship between a moth sex pheromone signal and fitness, and (ii) finding this relationship not only for the amount of pheromone produced, but also for the composition and for the extent which females keep their signal amount and composition stable. These results add multiple new dimensions in which genetic quality differences between female moths can contribute to pheromone variation within species. As genetic variation underlying sexual signals ultimately provides the raw material from which species barriers may originate, the relationship between signal variation and fitness components presented here thus provides a mechanism for the evolution of sex pheromone signals and the diversity of moth species.

Data accessibility. The data and scripts corresponding to this paper are archived online in the Dryad Digital Repository: https://doi.org/10.5061/dryad.6djh9w10f [77].

Authors' contributions. T.B., R.L. and A.T.G. conceived the study. R.L. and C.P. performed the experiments. D.v.V. supported animal breeding and helped with the experiments. T.B., R.L. and M.v.W. conducted the analysis. T.B. and A.T.G. wrote the manuscript with input from all other authors

Competing interests. We declare we have no competing interests.

Funding. This project was funded by the Netherlands Organization for Scientific Research (NWO-ALW, award no. 822.01.012 awarded to A.T.G.), the National Science Foundation (award nos IOS-1052238 and IOS-1456973) and by the Marie Skłodowska-Curie Individual fellowship (grant agreement no. 794254 awarded to T.B.).

Acknowledgements. The authors thank the Evolutionary and Population Biology group members, especially Isabel Smallegange and Emily Burdfield-Steel for their helpful discussions and the editorial board members and referees for their constructive comments and feedback.

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
