## [Peer Review File · Royal Society Open Science]

Review History

RSOS-210180.R0 (Original submission)

Review form: Reviewer 1

Is the manuscript scientifically sound in its present form?

Yes

Are the interpretations and conclusions justified by the results?

No

Is the language acceptable?

Yes

Do you have any ethical concerns with this paper?

No

Have you any concerns about statistical analyses in this paper?

No

Recommendation?

Major revision is needed (please make suggestions in comments)

Comments to the Author(s)

This study shows a covariant pattern of female fitness and pheromone production in a moth. The result is, in simple, that a female with a better fitness quality produce better pheromone. This is not surprising, and I think that the statements of the authors are generally rational.

But I found several issues to be addressed before considering publication. For example, the authors described that "We thus conclude that our results meet expectations for costly pheromone signaling (L347)", but I feel such the conclusion may misreading since the present study does not designed directly to assess the cost of pheromone signaling. I agree that "If sex pheromone signals are costly, calling activity, pheromone amount and/or pheromone composition should covary with fitness (L336)", but it is not always true that "if we found a covariation between pheromone production and fitness, pheromone signals are costly". The present study only shows the evidence of covariation between pheromone production and fitness, and I think the manuscript should be tone down more carefully; e.g. "In summary, we find evidence for signal-fitness covariation, which thus indicate costs to sex pheromone signaling (L428)" should be revised to "In summary, we find evidence for signal-fitness covariation, which may imply costs to sex pheromone signaling (L428)".

Moreover, trade-offs of investment to signaling and fitness are hard to assess by comparing 3-d-old and 8-d-old females, because it is difficult to exclude some effects of simple reproductive senescence. As mentioned in Introduction (L94), experiments in 3-d-old and 8-d-old females can only strengthen robustness of the authors' findings.

Other comments are as follows;

L73

the sex pheromone of > 2000 moth species has been identified

->

the sex pheromones and attractants of > 2000 moth species has been studied

[According to the latest list

(https://lepipheromone.sakura.ne.jp/PheromoneList/List_of_Sex_Pheromones_in_English_20210108.pdf), pheromones of ~700 species were identified. In the other ~1300 species, the findings only for attractants are reported.]

L133

important for male attraction (Z11-16:Ald, Z11-16:OAc, Z11-16:OH, and [Z7-16:Ald + Z9-16:Ald]), as well as the total amount of pheromone

->

important for male attraction; (Z)-11-hexadecenal (Z11-16:Ald), (Z)-11-hexadecenyl acetate (Z11-16:OAc), (Z)-11-hexadecenol (Z11-16:OH), and (Z)-7- and (Z)-9-hexadecenals (Z7-16:Ald + Z9-16:Ald), as well as the total amount of these components

[Descriptions such as "Z11-16:Ald" are only abbreviation in chemistry. IUPAC names should be shown when they are mentioned first.]

L422

Juvenile Hormone

->

juvenile hormone

Review form: Reviewer 2

Is the manuscript scientifically sound in its present form?

Yes

Are the interpretations and conclusions justified by the results?

Yes

Is the language acceptable?

Yes

Do you have any ethical concerns with this paper?

No

Have you any concerns about statistical analyses in this paper?

No

Recommendation?

Accept with minor revision (please list in comments)

Comments to the Author(s)

This manuscript describes analysis of young and old virgin female moth sex pheromone on the pheromone gland of *H. subflexa*. The study tries to demonstrate that there is a cost to producing pheromone and indicates that there is correlation between fitness and sex pheromone composition. It tries to explain why there can be variation in sex pheromone ratios in a population. Previous reviewers comments have been addressed appropriately. There will still be debate about whether or not that the observed variation in the pheromone signal will have biological relevance. A demonstration that male moths prefer younger females over older females would help establish the biological relevance. This could be done in a wind tunnel or an olfactometer. I would definitely like to see the absolute amounts of each sex pheromone component presented as nanogram values. This could be done in a supplementary table for all the females. It would greatly complement the statistics presented in the paper. The readers can then use these values to make their own conclusions. The same for the pupal weights.

Some specific comments:

Line 249 – trouble or double?

Line 299 – In figure 4 why are there differently colored circles in the early maters?

Line 378 – the statement that pheromone production does not have significant metabolic costs is not accurate. It has been shown that females with access to a nectar source (sugar) will produce more pheromone (Foster 2009, *J Exp Biol* 212:2789; Zhang et al. 2021, *Front Physiol* 11:605145). Therefore the nutrients used for sex pheromone biosynthesis are critical and if the female doesn't nectar feed she will have a lower pheromone amount.

Line 405 – factor of 2?

Review form: Reviewer 3

Is the manuscript scientifically sound in its present form?

No

Are the interpretations and conclusions justified by the results?

No

Is the language acceptable?

Yes

Do you have any ethical concerns with this paper?

No

Have you any concerns about statistical analyses in this paper?

No

Recommendation?

Major revision is needed (please make suggestions in comments)

Comments to the Author(s)

Sex pheromone signal and stability covary with fitness

RSOS-210180

Blankers et al.

I find the paper interesting and timely. The idea of sex pheromone in moths as a sexual trait and the relative importance of natural and sexual selection pressures in regulating sex pheromone variation is still strongly debated (Allison and Cardé 2016) and detailed experimental studies, as the current study, are important contributions.

In this study of *Heliothis subflexa*, two hypotheses were tested: Pheromone characteristics covary with fitness parameters, and (2) the maintenance of the signal is costly.

The pheromone was measured twice in the same individual, first on day one and second on day 3 or 8. Fecundity, longevity, and calling behavior were measured for each female. These allow for testing the associations among the three fitness parameters and 4 characteristics of the pheromone. The MS has an interesting experimental approach to question the link between pheromone characteristics and female fitness, it is well written and to my opinion and suits to be published in RSOS.

In the summary of the results the authors claim: 1) The delay in mating and continued investment in signaling was associated with low reproductive output, 2) Signal of females has changed with time, 3) some females maintained stable signal over time but in others the signal was changed with time, 4) heritability estimates of the variation in pheromone is low, 5) longevity, fecundity and fertility were correlated with pheromone characteristics and stability. They concluded that the results meet the expectation that pheromone signals are costly, and depend on the genetic quality of the females.

I have some reservations about this conclusion and other major comments

1. The study is using correlations and associations, no where in the study the cost of pheromone production or maintenance was measured directly or indirectly.
2. Along the paper conclusions concerning the effect of calling time are not tested against the effect of aging. Although it is difficult to separate the two, some manipulations can be done to prevent female of calling or to exhortate calling, thus to uncouple time of calling and aging.
3. Similarly, the effect of the heritability of the pheromone (although very low) is not tested against the effect of the heritability of size which is known to strongly affect pheromone amount and ratio.
4. Stability of the pheromone over time. Testing the stability of the pheromone characteristic emitted by the same female at different points of time is a beautiful approach. Keeping the pheromone characteristics in a deteriorating condition (age) may suggest (1) it is not condition dependent (2) it is condition dependent but only females in good condition can

maintain the pheromone characteristics. However, no information that links stability and female size is provided, and therefore many questions remained open. If small females (less fecund) can maintain pheromone stability, and stability, regardless of the pheromone characteristics are associated with higher fecundity, what does it signal for males? As males do not measure stability but may measure amount and ratio. Stability was related to fitness in early maters but not in old maters. Stability is easier to maintain between 3 days than between 8 days. How many females retained stability when are old compared to young? The results actually suggest that something else affected fitness with no relation to pheromone stability, such as age. Age is known to affect pheromone characteristics, thus aging results in low stability. Thus, in the end, stability did not contribute to our understanding of the link between pheromone characteristics and female fitness. Age is also known to affect fitness (and pheromone characteristics). Now, what left to close the circle is to find a correlation between pheromone characteristics and fitness, which was the aim of this study.

Back to cost

In order to demonstrate a cost of the pheromone as a trait, either the synthesis of the pheromone or energy needed for calling behavior, a different cost to females in good conditions and females in bad conditions should be apparent, such that the cost to females in bad condition is higher than that to females in good conditions. However, in this study samples of pheromone that were <10ng were excluded from the analyses and so were small individuals. By such doing, the mere essence of cost cannot be calculated (Zahavi 1975, 1977). If this is not the case, theoretically, cheating is an option; the signal does not represent the real condition of the individual and is losing its reliability. However, signals can still be honest and reliable without a cost (index signals vs. Maynard Smith and Harper 1995).

The study claims " Females that spent more nights calling had reduced reproductive success (line 357), and "these findings support our hypothesis that there are costs associated with sex pheromone calling in female moths (line 361-362). This sentence leans on correlation but not on causation. First, it may be true that the females experienced more days of calling and reduced fecundity but there is no evidence that the one had led to the other. Both are the effect of aging before mating. Second, no cost of calling was demonstrated.

Furthermore, Foster et al. 2018 demonstrated that in contrast to the assumption that costs of pheromone synthesis may limit the quantity of pheromone released (Harari et al. 2011; Johansson and Jones 2007; Umbers et al. 2015), that most pheromone synthesized is actually catabolized in the gland." Thus the cost of synthesis is neglected. On the other hand, this study found a limited correlation between calling duration and fitness parameters, thus the cost of calling behavior is limited too. These arguments have to be discussed.

This however does not exclude pheromone characteristics from signaling the female quality.

Pheromone characteristic may provide honest information about the female fecundity with no apparent cost, as an index (Maynard Smith and Harper 1995 and others).

This brings me to the next problem I have with the paper, calling activity.

Calling activity, or calling behavior was not defined. However, this was observed for each female every night - what does that mean? What parameters taken?

Per individual onset of calling and calling activity in a time leg of 30 min (for how long?) is problematic, especially if females call intermittently. Calling once in 30 min or 30 times in 30 min are scored the same. The two, however, may have a different cost. So the cost of calling cannot be calculated or assumed from the results. In addition, there is a correlation between late maters (many days of calling) and increasing life span. This contradicts the conclusion that calling behavior is costly.

Indeed it seems that the variance in all parameters related to calling activity precluded it from having a significant effect in the model ("as it explained little to no variation in fitness").

Therefore, the conclusion that high calling activity is related to reduced reproductive success or not may be misleading.

Calling time vs time (aging)

Along the way, (e.g. line 338) there is no separation between calling time and aging, as both are increased with time. Thus, the suggested effect of calling time on fitness may also be the effect of aging on fitness. There is no evidence that continued investment in signaling has an effect on reproductive output, this association may most probably be mediated through aging. Thus, although pheromone of aging females is highly associated with female fitness at this age, it does not have to be the effect of many days of calling.

Calling behavior and female mass

Since the study is based on correlations and not causations, it is difficult to tell apart the causes from the effects. For example, in line 275 calling behavior and pupal mass explain fitness. There is a well-known effect of female size on fecundity with no relation to pheromone amount or calling time. So, was the effect due to female size or calling time?

For example, from the results – more pheromone, or higher relative amount of each of the 3 main components were correlated with higher fecundity, fertility, and life span.

Large pupae have more OH and more eggs (line 293)

Small pupae have more OH and fewer eggs. This suggests the effect of size but not of OH.

Larger pupae have more OAc and a longer lifespan

Small pupae have more OAc but a shorter lifespan. This suggests the effect of size but not of OAc.

Minor comments

Abstract

Line 18 – "...continue the maintenance of genetic variation in a signal trait, despite selection from mate preference". This sentence is misleading, selection for mate preference (sexual selection) actually maintains high variance in the trait (the lek paradox).

Line 29 – in both groups, we found.... Longevity to be correlated with pheromone amount- However, for early maters in negatively correlated, and for late maters I negatively correlated. Should be emphasized.

Line 30 "This study is the first to report a correlation between fitness and sex pheromone composition in moths. This is a misleading sentence as "correlation between fitness and sex pheromone composition in moths" was already reported in Harari, Zahavi, Thiéry 2011 (#39 in the reference list).

Line 43 "additionally, in many organisms sexual signals are used to localize potentially suitable conspecific mates". I presume you mean here assortative mating, choosing among conspecific or natural selection choosing conspecifics from interspecifics. It is not clear from the sentence. Nevertheless, in both cases, a citation is missing. The following sentence is not clear either, what are these so-called species recognition signals, those you defined earlier (line 42) as under directional selection? Or those used for "suitable conspecific mates" which are also under sexual selection (although not directional). This paragraph needs clarification.

Line 65 –change to - see review in 30).

Line 76 there are not many papers in moths but quite a few in chemical communication in mammals (See review in Penn 2002 or Boulet et al. 2010) and in other insects (for example Rantala et al. 2003, Moore et al. 1995, 1997; Worden et al. 2000; Chemnitz et al. 2015; and in moths - Jaffe et al. 2007; Harari et al. 2011). Please clarify

Line 83 – signaling activity is mentioned here for the first time but it was not defined anywhere and not how was it measured. Please clarify.

Line 106. Emergence was checked mostly during the scotophase. Do adults not emerge in the photophase? Or was it due to convenience working hours? Please clarify.

Line 138 – Please explain why samples containing <10 ng are not reliable

Line 153 – to measure fecundity females were mated. How? Was there a possibility to choose a mate or you assigned a male to a female? This may affect the results. As females mate more than

once, young females that were assigned to low-quality males may delay oviposition, awaiting a better male.

What was the males' age? Was it similar in both early and late maters? Old males may have less fertilizing sperm. Please clarify.

Line 226. Old females lay fewer eggs than young females. The phenomenon is known. Here the assumption (line 226) is that the reduced fertility is due to loss of time – missing 5 days of laying, i.e. all females young and old are capable of laying X eggs/day. That is, females of old age do not have less energy or fewer viable eggs when are old. Is there any physiological support for this? Alternatively, a similar mean number of eggs per day of old and young females is only a coincidence, maybe due to the distribution of female size in the two groups? Please clarify

Line 325 – we also observed examples of effects in the opposite direction for both fertility and lifespan. What does that mean? How often?

Lines 382-390. The arguments in the paragraph are not clear. 1) High relative amount of Z11-16:OAc is correlated with higher fitness of females (Line 287). (2) Apparently, Z11-16:OAc is only produced when communication interference is likely to happen (line 388). (3) In lab colony, in the absence of *H. virescens* for many generations Z11-16:OAc not only exists but also associates with higher fitness (line 391). (4) A surprising conclusion – “selection likely favors females that produce lower relative amounts of acetate esters when there is no risk of heterospecific mate attraction”. So, is more of the Z11-16:OAc related to better fitness in the lab population (1) or not (4)? If pheromones are costly to produce, why Z11-16:OAc is still produced after so many generations in the lab in the absence of *H. virescens*? What is the relative ratio of this component in the pheromone blend? Please clarify.

Line 414. Stability is correlated with higher fitness, but a negative correlation was also found. This is puzzling. What ratio of the females had a positive effect on fitness and what ratio of negative effect?

Line 415. “The evidence thus supports our hypothesis for fitness costs to sex pheromone stability”. Again, the cost may be assumed but there is no evidence for the such.

Line 417. “it is possible that female benefits from maintaining a young-female-like blend”. This is an interesting argument and should be elaborated. Does this mean that females cheat regarding their age. Do they also cheat regarding their fecundity? Alternatively, males do not care about females' age but their mind their fecundity, thus females advertise their fecundity, not their age. But, did stability correlate also with size? This may strengthen the argument of pheromone as an honest signal, as only females with high fitness may keep signaling with their quality despite their aging. Please clarify.

431- the relationship between a moth sex pheromone signal and fitness is already studied in Harari et al. 2011. Please clarify.

Decision letter (RSOS-210180.R0)

Dear Mr Blankers

The Editors assigned to your paper RSOS-210180 "Sex pheromone signal and stability covary with fitness" have now received comments from reviewers and would like you to revise the paper in accordance with the reviewer comments and any comments from the Editors. Please note this decision does not guarantee eventual acceptance.

Please submit your revised manuscript and required files (see below) no later than 21 days from today's (ie 12-Apr-2021) date. Note: the ScholarOne system will 'lock' if submission of the revision is attempted 21 or more days after the deadline. If you do not think you will be able to meet this deadline please contact the editorial office immediately.

on behalf of Kevin Padian (Subject Editor)
openscience@royalsociety.org

Associate Editor Comments to Author:
Comments to the Author:

While there is clearly merit in further considering this paper, the three reviewers who have provided commentary each recommend a number of amendments that should be addressed in a revision. Please clearly delineate the changes you make in response to the referees in a new version of the paper. Good luck!

Reviewer comments to Author:
Reviewer: 1

Comments to the Author(s)

This study shows a covariant pattern of female fitness and pheromone production in a moth. The result is, in simple, that a female with a better fitness quality produce better pheromone. This is not surprising, and I think that the statements of the authors are generally rational.

But I found several issues to be addressed before considering publication. For example, the authors described that "We thus conclude that our results meet expectations for costly pheromone signaling (L347)", but I feel such the conclusion may misreading since the present study does not designed directly to assess the cost of pheromone signaling. I agree that "If sex pheromone signals are costly, calling activity, pheromone amount and/or pheromone composition should covary with fitness (L336)", but it is not always true that "if we found a

covariation between pheromone production and fitness, pheromone signals are costly". The present study only shows the evidence of covariation between pheromone production and fitness, and I think the manuscript should be tone down more carefully; e.g. "In summary, we find evidence for signal-fitness covariation, which thus indicate costs to sex pheromone signaling (L428)" should be revised to "In summary, we find evidence for signal-fitness covariation, which may imply costs to sex pheromone signaling (L428)".

Moreover, trade-offs of investment to signaling and fitness are hard to assess by comparing 3-d-old and 8-d-old females, because it is difficult to exclude some effects of simple reproductive senescence. As mentioned in Introduction (L94), experiments in 3-d-old and 8-d-old females can only strengthen robustness of the authors' findings.

Other comments are as follows;

L73

the sex pheromone of > 2000 moth species has been identified

->

the sex pheromones and attractants of > 2000 moth species has been studied

[According to the latest list

(https://lepipheromone.sakura.ne.jp/PheromoneList/List_of_Sex_Pheromones_in_English_20210108.pdf), pheromones of ~700 species were identified. In the other ~1300 species, the findings only for attractants are reported.]

L133

important for male attraction (Z11-16:Ald, Z11-16:OAc, Z11-16:OH, and [Z7-16:Ald + Z9-16:Ald]), as well as the total amount of pheromone

->

important for male attraction; (Z)-11-hexadecenal (Z11-16:Ald), (Z)-11-hexadecenyl acetate (Z11-16:OAc), (Z)-11-hexadecenol (Z11-16:OH), and (Z)-7- and (Z)-9-hexadecenals (Z7-16:Ald + Z9-16:Ald), as well as the total amount of these components

[Descriptions such as "Z11-16:Ald" are only abbreviation in chemistry. IUPAC names should be shown when they are mentioned first.]

L422

Juvenile Hormone

->

juvenile hormone

Reviewer: 2

Comments to the Author(s)

This manuscript describes analysis of young and old virgin female moth sex pheromone on the pheromone gland of *H. subflexa*. The study tries to demonstrate that there is a cost to producing pheromone and indicates that there is correlation between fitness and sex pheromone composition. It tries to explain why there can be variation in sex pheromone ratios in a population. Previous reviewers comments have been addressed appropriately. There will still be debate about whether or not that the observed variation in the pheromone signal will have biological relevance. A demonstration that male moths prefer younger females over older females would help establish the biological relevance. This could be done in a wind tunnel or an olfactometer. I would definitely like to see the absolute amounts of each sex pheromone component presented as nanogram values. This could be done in a supplementary table for all the females. It would greatly complement the statistics presented in the paper. The readers can then use these values to make their own conclusions. The same for the pupal weights.

Some specific comments:

Line 249 – trouble or double?

Line 299 – In figure 4 why are there differently colored circles in the early maters?

Line 378 – the statement that pheromone production does not have significant metabolic costs is not accurate. It has been shown that females with access to a nectar source (sugar) will produce more pheromone (Foster 2009, *J Exp Biol* 212:2789; Zhang et al. 2021, *Front Physiol* 11:605145).

Therefore the nutrients used for sex pheromone biosynthesis are critical and if the female doesn't nectar feed she will have a lower pheromone amount.

Line 405 – factor of 2?

Reviewer: 3

Comments to the Author(s)

Sex pheromone signal and stability covary with fitness

RSOS-210180

Blankers et al.

I find the paper interesting and timely. The idea of sex pheromone in moths as a sexual trait and the relative importance of natural and sexual selection pressures in regulating sex pheromone variation is still strongly debated (Allison and Cardé 2016) and detailed experimental studies, as the current study, are important contributions.

In this study of *Heliothis subflexa*, two hypotheses were tested: Pheromone characteristics covary with fitness parameters, and (2) the maintenance of the signal is costly.

The pheromone was measured twice in the same individual, first on day one and second on day 3 or 8. Fecundity, longevity, and calling behavior were measured for each female. These allow for testing the associations among the three fitness parameters and 4 characteristics of the pheromone. The MS has an interesting experimental approach to question the link between pheromone characteristics and female fitness, it is well written and to my opinion and suits to be published in RSOS.

In the summary of the results the authors claim: 1) The delay in mating and continued investment in signaling was associated with low reproductive output, 2) Signal of females has changed with time, 3) some females maintained stable signal over time but in others the signal was changed with time, 4) heritability estimates of the variation in pheromone is low, 5) longevity, fecundity and fertility were correlated with pheromone characteristics and stability. They concluded that the results meet the expectation that pheromone signals are costly, and depend on the genetic quality of the females.

I have some reservations about this conclusion and other major comments

1. The study is using correlations and associations, no where in the study the cost of pheromone production or maintenance was measured directly or indirectly.
2. Along the paper conclusions concerning the effect of calling time are not tested against the effect of aging. Although it is difficult to separate the two, some manipulations can be done to prevent female of calling or to exhilarate calling, thus to uncouple time of calling and aging.
3. Similarly, the effect of the heritability of the pheromone (although very low) is not tested against the effect of the heritability of size which is known to strongly affect pheromone amount and ratio.
4. Stability of the pheromone over time. Testing the stability of the pheromone characteristic emitted by the same female at different points of time is a beautiful approach. Keeping the pheromone characteristics in a deteriorating condition (age) may suggest (1) it is not condition dependent (2) it is condition dependent but only females in good condition can maintain the pheromone characteristics. However, no information that links stability and female size is

provided, and therefore many questions reminded open. If small females (less fecund) can maintain pheromone stability, and stability, regardless of the pheromone characteristics are associated with higher fecundity, what does it signal for males? As males do not measure stability but may measure amount and ratio. Stability was related to fitness in early maters but not in old maters. Stability is easier to maintain between 3 days than between 8 days. How many females retained stability when are old compared to young? The results actually suggest that something else affected fitness with no relation to pheromone stability, such as age. Age is known to affect pheromone characteristics, thus aging results in low stability. Thus, in the end, stability did not contribute to our understanding of the link between pheromone characteristics and female fitness. Age is also known to affect fitness (and pheromone characteristics). Now, what left to close the circle is to find a correlation between pheromone characteristics and fitness, which was the aim of this study.

Back to cost

In order to demonstrate a cost of the pheromone as a trait, either the synthesis of the pheromone or energy needed for calling behavior, a different cost to females in good conditions and females in bad conditions should be apparent, such that the cost to females in bad condition is higher than that to females in good conditions. However, in this study samples of pheromone that were <10ng were excluded from the analyses and so were small individuals. By such doing, the mere essence of cost cannot be calculated (Zahavi 1975, 1977). If this is not the case, theoretically, cheating is an option; the signal does not represent the real condition of the individual and is losing its reliability. However, signals can still be honest and reliable without a cost (index signals vs. Maynard Smith and Harper 1995).

The study claims " Females that spent more nights calling had reduced reproductive success (line 357), and "these findings support our hypothesis that there are costs associated with sex pheromone calling in female moths (line 361-362). This sentence leans on correlation but not on causation. First, it may be true that the females experienced more days of calling and reduced fecundity but there is no evidence that the one had led to the other. Both are the effect of aging before mating. Second, no cost of calling was demonstrated.

Furthermore, Foster et al. 2018 demonstrated that in contrast to the assumption that costs of pheromone synthesis may limit the quantity of pheromone released (Harari et al. 2011; Johansson and Jones 2007; Umbers et al. 2015), that most pheromone synthesized is actually catabolized in the gland." Thus the cost of synthesis is neglected. On the other hand, this study found a limited correlation between calling duration and fitness parameters, thus the cost of calling behavior is limited too. These arguments have to be discussed.

This however does not exclude pheromone characteristics from signaling the female quality. Pheromone characteristic may provide honest information about the female fecundity with no apparent cost, as an index (Maynard Smoth and Harper 1995 and others).

This brings me to the next problem I have with the paper, calling activity.

Calling activity, or calling behavior was not defined. However, this was observed for each female every night - what does that mean? What parameters taken?

Per individual onset of calling and calling activity in a time leg of 30 min (for how long?) is problematic, especially if females call intermittently. Calling once in 30 min or 30 times in 30 min are scored the same. The two, however, may have a different cost. So the cost of calling cannot be calculated or assumed from the results. In addition, there is a correlation between late maters (many days of calling) and increasing life span. This contradicts the conclusion that calling behavior is costly.

Indeed it seems that the variance in all parameters related to calling activity precluded it from having a significant effect in the model ("as it explained little to no variation in fitness").

Therefore, the conclusion that high calling activity is related to reduced reproductive success or not may be misleading.

Calling time vs time (aging)

Along the way, (e.g. line 338) there is no separation between calling time and aging, as both are increased with time. Thus, the suggested effect of calling time on fitness may also be the effect of aging on fitness. There is no evidence that continued investment in signaling has an effect on reproductive output, this association may most probably be mediated through aging. Thus, although pheromone of aging females is highly associated with female fitness at this age, it does not have to be the effect of many days of calling.

Calling behavior and female mass

Since the study is based on correlations and not causations, it is difficult to tell apart the causes from the effects. For example, in line 275 calling behavior and pupal mass explain fitness. There is a well-known effect of female size on fecundity with no relation to pheromone amount or calling time. So, was the effect due to female size or calling time?

For example, from the results – more pheromone, or higher relative amount of each of the 3 main components were correlated with higher fecundity, fertility, and life span.

Large pupae have more OH and more eggs (line 293)

Small pupae have more OH and fewer eggs. This suggests the effect of size but not of OH.

Larger pupae have more OAc and a longer lifespan

Small pupae have more OAc but a shorter lifespan. This suggests the effect of size but not of OAc.

Minor comments

Abstract

Line 18 – "...continue the maintenance of genetic variation in a signal trait, despite selection from mate preference". This sentence is misleading, selection for mate preference (sexual selection) actually maintains high variance in the trait (the lek paradox).

Line 29 – in both groups, we found.... Longevity to be correlated with pheromone amount- However, for early maters in negatively correlated, and for late maters I negatively correlated. Should be emphasized.

Line 30 "This study is the first to report a correlation between fitness and sex pheromone composition in moths. This is a misleading sentence as "correlation between fitness and sex pheromone composition in moths" was already reported in Harari, Zahavi, Thiéry 2011 (#39 in the reference list).

Line 43 "additionally, in many organisms sexual signals are used to localize potentially suitable conspecific mates". I presume you mean here assortative mating, choosing among conspecific or natural selection choosing conspecifics from interspecifics. It is not clear from the sentence.

Nevertheless, in both cases, a citation is missing. The following sentence is not clear either, what are these so-called species recognition signals, those you defined earlier (line 42) as under directional selection? Or those used for "suitable conspecific mates" which are also under sexual selection (although not directional). This paragraph needs clarification.

Line 65 –change to - see review in 30).

Line 76 there are not many papers in moths but quite a few in chemical communication in mammals (See review in Penn 2002 or Boulet et al. 2010) and in other insects (for example Rantala et al. 2003, Moore et al. 1995, 1997; Worden et al. 2000; Chemnitz et al. 2015; and in moths - Jaffe et al. 2007; Harari et al. 2011). Please clarify

Line 83 – signaling activity is mentioned here for the first time but it was not defined anywhere and not how was it measured. Please clarify.

Line 106. Emergence was checked mostly during the scotophase. Do adults not emerge in the photophase? Or was it due to convenience working hours? Please clarify.

Line 138 – Please explain why samples containing <10 ng are not reliable

Line 153 – to measure fecundity females were mated. How? Was there a possibility to choose a mate or you assigned a male to a female? This may affect the results. As females mate more than once, young females that were assigned to low-quality males may delay oviposition, awaiting a better male.

What was the males' age? Was it similar in both early and late maters? Old males may have less fertilizing sperm. Please clarify.

Line 226. Old females lay fewer eggs than young females. The phenomenon is known. Here the assumption (line 226) is that the reduced fertility is due to loss of time – missing 5 days of laying, i.e. all females young and old are capable of laying X eggs/day. That is, females of old age do not have less energy or fewer viable eggs when are old. Is there any physiological support for this? Alternatively, a similar mean number of eggs per day of old and young females is only a coincidence, maybe due to the distribution of female size in the two groups? Please clarify

Line 325 – we also observed examples of effects in the opposite direction for both fertility and lifespan. What does that mean? How often?

Lines 382-390. The arguments in the paragraph are not clear. 1) High relative amount of Z11-16:OAc is correlated with higher fitness of females (Line 287). (2) Apparently, Z11-16:OAc is only produced when communication interference is likely to happen (line 388). (3) In lab colony, in the absence of *H. virescens* for many generations Z11-16:OAc not only exists but also associates with higher fitness (line 391). (4) A surprising conclusion – “selection likely favors females that produce lower relative amounts of acetate esters when there is no risk of heterospecific mate attraction”. So, is more of the Z11-16:OAc related to better fitness in the lab population (1) or not (4)? If pheromones are costly to produce, why Z11-16:OAc is still produced after so many generations in the lab in the absence of *H. virescens*? What is the relative ratio of this component in the pheromone blend? Please clarify.

Line 414. Stability is correlated with higher fitness, but a negative correlation was also found. This is puzzling. What ratio of the females had a positive effect on fitness and what ratio of negative effect?

Line 415. “The evidence thus supports our hypothesis for fitness costs to sex pheromone stability”. Again, the cost may be assumed but there is no evidence for the such.

Line 417. “it is possible that female benefits from maintaining a young-female-like blend”. This is an interesting argument and should be elaborated. Does this mean that females cheat regarding their age. Do they also cheat regarding their fecundity? Alternatively, males do not care about females' age but their mind their fecundity, thus females advertise their fecundity, not their age. But, did stability correlate also with size? This may strengthen the argument of pheromone as an honest signal, as only females with high fitness may keep signaling with their quality despite their aging. Please clarify.

431- the relationship between a moth sex pheromone signal and fitness is already studied in Harari et al. 2011. Please clarify.

===PREPARING YOUR MANUSCRIPT===

===PREPARING YOUR REVISION IN SCHOLARONE===

<https://royalsociety.org/journals/authors/author-guidelines/#data>. You should ensure that

you cite the dataset in your reference list. If you have deposited data etc in the Dryad repository, please include both the 'For publication' link and 'For review' link at this stage.

Author's Response to Decision Letter for (RSOS-210180.R0)

See Appendix A.

RSOS-210180.R1 (Revision)

Review form: Reviewer 1

Is the manuscript scientifically sound in its present form?

Yes

Are the interpretations and conclusions justified by the results?

Yes

Is the language acceptable?

Yes

Do you have any ethical concerns with this paper?

No

Have you any concerns about statistical analyses in this paper?

No

Recommendation?

Accept with minor revision (please list in comments)

Comments to the Author(s)

I feel the manuscript is improved following the reviewers' comments. I would like to give only an additional request; please add citation(s) of moth pheromone databases, such as the pherobase (<https://www.pherobase.com>) or the List of Pheromones and Attractants (<https://lepipheromone.sakura.ne.jp>) for your description in L69-70.

Review form: Reviewer 2

Is the manuscript scientifically sound in its present form?

Yes

Are the interpretations and conclusions justified by the results?

Yes

Is the language acceptable?

Yes

Do you have any ethical concerns with this paper?

No

Have you any concerns about statistical analyses in this paper?

No

Recommendation?

Accept as is

Comments to the Author(s)

In my opinion this manuscript can now be accepted for publication.

Review form: Reviewer 3

Is the manuscript scientifically sound in its present form?

Yes

Are the interpretations and conclusions justified by the results?

Yes

Is the language acceptable?

Yes

Do you have any ethical concerns with this paper?

No

Have you any concerns about statistical analyses in this paper?

No

Recommendation?

Accept with minor revision (please list in comments)

Comments to the Author(s)

Overall I am satisfied with the correction made on the MS.

However, 2 minor points remained to be addressed.

Line 152-153. : females were mated in 475-mL observation cups with virgin 2-3-

153 day-old males". Change to each female was mated in a 475-mL observation cup with a randomly chosen, virgin 2-3-day-old male..." or similar

Line 375 - "...it is possible that a female benefits from maintaining a young-female-like blend. This would be similar to... (72,73)." "This", here, refers to the possibility that old females benefit from maintaining a young-female-like blend. However, this is not what the two papers discuss. Rephrase.

Decision letter (RSOS-210180.R1)

Dear Mr Blankers

On behalf of the Editors, we are pleased to inform you that your Manuscript RSOS-210180.R1 "Sex pheromone signal and stability covary with fitness" has been accepted for publication in Royal Society Open Science subject to minor revision in accordance with the referees' reports. Please find the referees' comments along with any feedback from the Editors below my signature.

Please submit your revised manuscript and required files (see below) no later than 7 days from today's (ie 28-May-2021) date. Note: the ScholarOne system will 'lock' if submission of the revision is attempted 7 or more days after the deadline. If you do not think you will be able to meet this deadline please contact the editorial office immediately.

on behalf of Kevin Padian (Subject Editor)
openscience@royalsociety.org

Associate Editor Comments to Author:

Comments to the Author:

A few small (but useful) tweaks remain to be made, but overall, the reviewers are satisfied the paper is near ready for publication. Good job! We'll look forward to receiving your final version soon.

Reviewer comments to Author:

Reviewer: 2

Comments to the Author(s)

In my opinion this manuscript can now be accepted for publication.

Reviewer: 1

Comments to the Author(s)

I feel the manuscript is improved following the reviewers' comments. I would like to give only an additional request; please add citation(s) of moth pheromone databases, such as the pherobase (<https://www.pherobase.com>) or the List of Pheromones and Attractants (<https://lepipheromone.sakura.ne.jp>) for your description in L69-70.

Reviewer: 3

Comments to the Author(s)

Overall I am satisfied with the correction made on the MS.

However, 2 minor points remained to be addressed.

Line 152-153. : females were mated in 475-mL observation cups with virgin 2-3-153 day-old males". Change to each female was mated in a 475-mL observation cup with a randomly chosen, virgin 2-3-day-old male..." or similar

Line 375 - "...it is possible that a female benefits from maintaining a young-female-like blend. This would be similar to... (72,73)." "This", here, refers to the possibility that old females benefit from maintaining a young-female-like blend. However, this is not what the two papers discuss. Rephrase.

===PREPARING YOUR MANUSCRIPT===

While not essential, it will speed up the preparation of your manuscript proof if you format your references/bibliography in Vancouver style (please see

<https://royalsociety.org/journals/authors/author-guidelines/#formatting>). You should include DOIs for as many of the references as possible.

===PREPARING YOUR REVISION IN SCHOLARONE===

<https://royalsociety.org/journals/authors/author-guidelines/#data>. You should ensure that you cite the dataset in your reference list. If you have deposited data etc in the Dryad repository,

please only include the 'For publication' link at this stage. You should remove the 'For review' link.

Author's Response to Decision Letter for (RSOS-210180.R1)

See Appendix B.

Decision letter (RSOS-210180.R2)

Dear Mr Blankers,

I am pleased to inform you that your manuscript entitled "Sex pheromone signal and stability covary with fitness" is now accepted for publication in Royal Society Open Science.

Please see the Royal Society Publishing guidance on how you may share your accepted author manuscript at <https://royalsociety.org/journals/ethics-policies/media-embargo/>. After publication, some additional ways to effectively promote your article can also be found here

<https://royalsociety.org/blog/2020/07/promoting-your-latest-paper-and-tracking-your-results/>.

on behalf of Prof Kevin Padian (Subject Editor)
openscience@royalsociety.org

Appendix A

Dear Editor,

Please find enclosed the revised manuscript RSOS-210180 "Sex pheromone signal and stability covary with fitness". We appreciate the effort by the reviewers to improve the manuscript and we have addressed all comments below (blue font). Following their suggestions, we have added some interesting discussion points. In some locations we made changes to the text to more accurately reflect the confidence we can have in our conclusions. We also included a new supplementary table with the raw, absolute pheromone amounts and pupal weights. Please find our detailed responses to all comments below.

With these changes and answers to all comments, we hope that our manuscript can now be accepted for publication on Royal Society Open Science and we are looking forward to hear from you. In case you have any questions, please do not hesitate to contact me.

On behalf of all co-authors and with kind regards,

Thomas Blankers
University of Amsterdam

----- Point-wise response below in blue font -----

NB. All line numbers refer to the document with tracked changes

Associate Editor Comments to Author:

Comments to the Author:

While there is clearly merit in further considering this paper, the three reviewers who have provided commentary each recommend a number of amendments that should be addressed in a revision. Please clearly delineate the changes you make in response to the referees in a new version of the paper. Good luck!

Reviewer comments to Author:

Reviewer: 1

Comments to the Author(s)

This study shows a covariant pattern of female fitness and pheromone production in a moth. The result is, in simple, that a female with a better fitness quality produce better pheromone. This is not surprising, and I think that the statements of the authors are generally rational.

But I found several issues to be addressed before considering publication. For example, the authors described that “We thus conclude that our results meet expectations for costly pheromone signaling (L347)”, but I feel such the conclusion may misreading since the present study does not designed directly to assess the cost of pheromone signaling. I agree that “If sex pheromone signals are costly, calling activity, pheromone amount and/or pheromone composition should covary with fitness (L336)”, but it is not always true that “if we found a covariation between pheromone production and fitness, pheromone signals are costly”. The present study only shows the evidence of covariation between pheromone production and fitness, and I think the manuscript should be tone down more carefully; e.g. “In summary, we find evidence for signal-fitness covariation, which thus indicate costs to sex pheromone signaling (L428)” should be revised to “In summary, we find evidence for signal-fitness covariation, which may imply costs to sex pheromone signaling (L428)”.

We agree that our conclusions should not suggest that they were based on direct measurements of costs. We have modified the specific example in Ln 428 (ln 447) according to the reviewer’s suggestion as well as one other instance in the discussion (ln 386), where we replaced “costly sex pheromone signals” with “signal-fitness covariance”.

Moreover, trade-offs of investment to signaling and fitness are hard to assess by comparing 3-d-old and 8-d-old females, because it is difficult to exclude some effects of simple reproductive senescence. As mentioned in Introduction (L94), experiments in 3-d-old and 8-d-old females can only strengthen robustness of the authors’ findings.

The reviewer is right that reproductive senescence indeed could be a confounding factor that we cannot exclude. We now modified the end of the discussion paragraph as follows: “These findings are in line with general patterns across moths (60) and suggest that the additional days spent calling in the slightly longer-lived late maters reduce the energy and time available to lay eggs, implying costs to calling. However, we acknowledge that there may be confounding effects from reproductive senescence on the fitness differences between early and late maters” (ln 375)

Other comments are as follows;

L73

the sex pheromone of > 2000 moth species has been identified

->

the sex pheromones and attractants of > 2000 moth species has been studied

[According to the latest list

(https://lepipheromone.sakura.ne.jp/PheromoneList/List_of_Sex_Pheromones_in_English_20210108.pdf), pheromones of ~700 species were identified. In the other ~1300 species, the findings only for attractants are reported.]

changed accordingly

L133

important for male attraction (Z11-16:Ald, Z11-16:OAc, Z11-16:OH, and [Z7-16:Ald + Z9-16:Ald]), as well as the total amount of pheromone

->

important for male attraction; (Z)-11-hexadecenal (Z11-16:Ald), (Z)-11-hexadecenyl acetate (Z11-16:OAc), (Z)-11-hexadecenol (Z11-16:OH), and (Z)-7- and (Z)-9-hexadecenals (Z7-16:Ald + Z9-16:Ald), as well as the total amount of these components

[Descriptions such as “Z11-16:Ald” are only abbreviation in chemistry. IUPAC names should be shown when they are mentioned first.]

changed accordingly

L422

Juvenile Hormone

->

juvenile hormone

changed accordingly

Reviewer: 2

Comments to the Author(s)

This manuscript describes analysis of young and old virgin female moth sex pheromone on the pheromone gland of *H. subflexa*. The study tries to demonstrate that there is a cost to producing pheromone and indicates that there is correlation between fitness and sex pheromone composition. It tries to explain why there can be variation in sex pheromone ratios in a population. Previous reviewers comments have been addressed appropriately. There will still be debate about whether or not that the observed variation in the pheromone signal will have biological relevance. A demonstration that male moths prefer younger females over older females would help establish the biological relevance. This could be done in a wind tunnel or an olfactometer.

We refer to previous research showing that across moth species, older females are less attractive to males than younger females. We do so both in the introduction: “In moths, older (virgin)

females tend to have reduced mating activity and reduced mating success (42–48)” and in the discussion: “Since in moths older females are typically less attractive, as measured by mating success (42–48)”. Choice or response experiments are outside the scope of this paper.

I would definitely like to see the absolute amounts of each sex pheromone component presented as nanogram values. This could be done in a supplementary table for all the females. It would greatly complement the statistics presented in the paper. The readers can then use these values to make their own conclusions. The same for the pupal weights.

We appreciate the suggestion to include the raw pheromone/life history measurements as a supplementary table in addition to the archived data (where all raw data are available), which typically require readers more effort to access. We have therefore added a supplementary table (now Table S2) and a cross reference in the results section about pheromone variation (In 255).

Some specific comments:

Line 249 – trouble or double? We meant trouble, but see how this sentence is confusing, so changed “trouble the use of relative amounts” to “are problematic in the analysis of relative amounts”.

Line 299 – In figure 4 why are there differently colored circles in the early maters? Different color coding for different terciles in interactions

Line 378 – the statement that pheromone production does not have significant metabolic costs is not accurate. It has been shown that females with access to a nectar source (sugar) will produce more pheromone (Foster 2009, J Exp Biol 212:2789; Zhang et al. 2021, Front Physiol 11:605145). Therefore the nutrients used for sex pheromone biosynthesis are critical and if the female doesn't nectar feed she will have a lower pheromone amount.

We now added the following sentence to this discussion: “However, sugar feeding does increase pheromone titers (61,62) and starved females produce low titers and are less attractive (62).

Line 405 – factor of 2? Probably the confusion came from writing 2-fold instead of twofold? We now changed it to twofold.

Reviewer: 3

Comments to the Author(s)

Sex pheromone signal and stability covary with fitness

RSOS-210180

Blankers et al.

I find the paper interesting and timely. The idea of sex pheromone in moths as a sexual trait and the relative importance of natural and sexual selection pressures in regulating sex pheromone variation is still strongly debated (Allison and Cardé 2016) and detailed experimental studies, as the current study, are important contributions.

In this study of *Heliothis subflexa*, two hypotheses were tested: Pheromone characteristics covary with fitness parameters, and (2) the maintenance of the signal is costly.

The pheromone was measured twice in the same individual, first on day one and second on day 3 or 8. Fecundity, longevity, and calling behavior were measured for each female. These allow for testing the associations among the three fitness parameters and 4 characteristics of the pheromone. The MS has an interesting experimental approach to question the link between pheromone characteristics and female fitness, it is well written and to my opinion and suits to be published in RSOS.

In the summary of the results the authors claim: 1) The delay in mating and continued investment in signaling was associated with low reproductive output, 2) Signal of females has changed with time, 3) some females maintained stable signal over time but in others the signal was changed with time, 4) heritability estimates of the variation in pheromone is low, 5) longevity, fecundity and fertility were correlated with pheromone characteristics and stability. They concluded that the results meet the expectation that pheromone signals are costly, and depend on the genetic quality of the females.

I have some reservations about this conclusion and other major comments

1. The study is using correlations and associations, no where in the study the cost of pheromone production or maintenance was measured directly or indirectly.

This is correct and in response to this comment and R1 we have phrased some conclusions with more nuance to reflect this.

2. Along the paper conclusions concerning the effect of calling time are not tested against the effect of aging. Although it is difficult to separate the two, some manipulations can be done to prevent female of calling or to exhilarate calling, thus to uncouple time of calling and aging.

The main confounding factor in here is reproductive senescence, also mentioned by R1, which we now acknowledge in the Discussion.

3. Similarly, the effect of the heritability of the pheromone (although very low) is not tested against the effect of the heritability of size which is known to strongly affect pheromone amount and ratio.

We do not understand what the reviewer means. We did not test the effect of heritability, we only measured heritability to provide context to the evolutionary relevance of signal and signal stability variance. Similarly, we did not measure the heritability of size. We did measure the size (pupal mass) of the phenotyped individuals and always included pupal mass as a covariate in the analyses.

4. Stability of the pheromone over time. Testing the stability of the pheromone characteristic emitted by the same female at different points of time is a beautiful approach. Keeping the pheromone characteristics in a deteriorating condition (age) may suggest (1) it is not condition dependent (2) it is condition dependent but only females in good condition can maintain the pheromone characteristics. However, no information that links stability and female size is provided, and therefore many questions remained open.

We respectfully disagree with the reviewer, because size (pupal mass) is always a covariate in our model relating stability to fitness, so that the predictive value of stability for fitness variation is that of the residuals of pheromone stability regressed on pupal mass. We also did not find any correlation between pupal mass and signal stability in our data, but since this not relevant to the study, we did not report this in the manuscript.

If small females (less fecund) can maintain pheromone stability, and stability, regardless of the pheromone characteristics are associated with higher fecundity, what does it signal for males? As males do not measure stability but may measure amount and ratio.

Since our manuscript focuses specifically on the effects of female fitness on the pheromone stability, and not on the male response, we do not discuss stability in the context of what it signals to males.

Stability was related to fitness in early maters but not in old maters.

Stability was related to fitness measurements in both early and late maters, except for the stability of total amount (stability in PC1), which was indeed uncoupled in late maters.

Stability is easier to maintain between 3 days than between 8 days. How many females retained stability when are old compared to young? The results actually suggest that something else affected fitness with no relation to pheromone stability, such as age.

Unfortunately, we do not follow the reviewer here. Specifically, what does the reviewer mean with “maintaining” or “retaining” stability? The fact that our findings are robust across the early and late maters argues against undue effects from age (which here again probably means reproductive senescence). Besides, age is not a factor in this analysis, because the fitness measured is over the entire life time of a female, and within early or late maters, all females were equally old (i.e. either 4 or 9 days) when mated.

Age is known to affect pheromone characteristics, thus aging results in low stability. Thus, in the end, stability did not contribute to our understanding of the link between pheromone characteristics and female fitness. Age is also known to affect fitness (and pheromone

characteristics). Now, what left to close the circle is to find a correlation between pheromone characteristics and fitness, which was the aim of this study.

Unfortunately, we again do not follow the reviewer. Age is not typically seen as something to affect fitness, as fitness is a life time total. Maybe the reviewer means that there is an effect from reproductive senescence when females are mated later in life? This we acknowledge in the text (line 375). Aging also does not result in low stability. Over time, as females age (as virgins at least), the signal changes linearly with time. We do not see an overall difference in the coefficient of this change if we compare early and late mated females and there is no a priori effect from mating delay on the stability. The condition-dependent stability contributes to our understanding of the link between pheromone and fitness in the following way. We found that the pheromone signal prior to mating predicts fitness and this effect is both in the early and late maters. In addition, we found that how much the signal changes over the life time of a female is also correlated with fitness. Therefore, we conclude that the pheromone signal in this moth is likely informative of her fitness across her entire life, even though the signal changes due to aging.

Back to cost

In order to demonstrate a cost of the pheromone as a trait, either the synthesis of the pheromone or energy needed for calling behavior, a different cost to females in good conditions and females in bad conditions should be apparent, such that the cost to females in bad condition is higher than that to females in good conditions. However, in this study samples of pheromone that were <10ng were excluded from the analyses and so were small individuals. By such doing, the mere essence of cost cannot be calculated (Zahavi 1975, 1977). If this is not the case, theoretically, cheating is an option; the signal does not represent the real condition of the individual and is losing its reliability. However, signals can still be honest and reliable without a cost (index signals vs. Maynard Smith and Harper 1995).

The reason of excluding pheromone samples < 10 ng is technical: in contrast to many other studies on pheromone composition that focus only on the major sex pheromone component, we integrate the whole blend, including the secondary sex pheromone components that are present at a few percent of the major component. Therefore we cannot include integrations from total amounts < 10 ng, because the data analysis is not reliable at such low amounts. The minimum weight for pupae used reflects a threshold that only included dead pupae. Therefore, these pupae were discarded immediately after measuring, because we knew that no life females were going to emerge from these pupae.

The study claims “ Females that spent more nights calling had reduced reproductive success (line 357), and “these findings support our hypothesis that there are costs associated with sex pheromone calling in female moths (line 361-362). This sentence leans on correlation but not on causation. First, it may be true that the females experienced more days of calling and reduced fecundity but there is no evidence that the one had led to the other. Both are the effect of aging before mating. Second, no cost of calling was demonstrated.

We now changed these sentences, also in response to comments from R1, see line 367-377

Furthermore, Foster et al. 2018 demonstrated that in contrast to the assumption that costs of pheromone synthesis may limit the quantity of pheromone released (Harari et al. 2011; Johansson and Jones 2007; Umbers et al. 2015), that most pheromone synthesized is actually catabolized in the gland.” Thus the cost of synthesis is neglected. On the other hand, this study found a limited correlation between calling duration and fitness parameters, thus the cost of calling behavior is limited too. These arguments have to be discussed.

The reviewer is right. We now added “...and because most the storage of release of pheromone titers is not constrained by the level of synthesis, but by the level of breakdown (catabolism) (Foster et al. 2018)” to the discussion of why we do not necessarily expect significant costs to pheromone production. We appreciate the suggestion, because R2 actually suggested literature that supports costs to the synthesis. So thanks to R2 and R3, this discussion paragraph now nicely reflects the uncertainty: on the one hand, nutrients used for synthesis are negligible relative to feeding rates and catabolism outweighs signaling. On the other hand, sugar feeding (compared to water feeding) increases pheromone titers to the extent that females become more attractive to males.

This however does not exclude pheromone characteristics from signaling the female quality. Pheromone characteristic may provide honest information about the female fecundity with no apparent cost, as an index (Maynard Smoth and Harper 1995 and others).

This brings me to the next problem I have with the paper, calling activity.

Calling activity, or calling behavior was not defined. However, this was observed for each female every night - what does that mean? What parameters taken?

Per individual onset of calling and calling activity in a time leg of 30 min (for how long?) is problematic, especially if females call intermittently. Calling once in 30 min or 30 times in 30 min are scored the same. The two, however, may have a different cost. So the cost of calling cannot be calculated or assumed from the results. In addition, there is a correlation between late maters (many days of calling) and increasing life span. This contradicts the conclusion that calling behavior is costly.

Indeed it seems that the variance in all parameters related to calling activity precluded it from having a significant effect in the model (“as it explained little to no variation in fitness”). Therefore, the conclusion that high calling activity is related to reduced reproductive success or not may be misleading.

Every observation of calling translates to a 30-minute calling duration, because females of this species and in our setup call almost continuously with only brief interruptions, as we have observed this over the past 15 years with many student projects. Therefore, we find it unlikely that our protocol results in significant overestimates of the true calling duration.

It is not contradictory when analyses within early or late maters suggest time spent calling has a smaller (or no) effect on fitness compared to analyses that compare early and late maters. This is

because there is much more variation in time spent calling between versus within groups, as late maters spent an additional 5 days calling (with > 80% of the females calling). There are also differences in how the analyses are set up, with the most important difference being the confounding effect of reproductive senescence in the comparison of early and late maters. This is now acknowledged in the Discussion (ln 375; see also our response to R1).

Calling time vs time (aging)

Along the way, (e.g. line 338) there is no separation between calling time and aging, as both are increased with time. Thus, the suggested effect of calling time on fitness may also be the effect of aging on fitness. There is no evidence that continued investment in signaling has an effect on reproductive output, this association may most probably be mediated through aging. Thus, although pheromone of aging females is highly associated with female fitness at this age, it does not have to be the effect of many days of calling.

Please see our answer above and also in response to R1.

Calling behavior and female mass

Since the study is based on correlations and not causations, it is difficult to tell apart the causes from the effects. For example, in line 275 calling behavior and pupal mass explain fitness. There is a well-known effect of female size on fecundity with no relation to pheromone amount or calling time. So, was the effect due to female size or calling time?

For example, from the results – more pheromone, or higher relative amount of each of the 3 main components were correlated with higher fecundity, fertility, and life span.

Large pupae have more OH and more eggs (line 293)

Small pupae have more OH and fewer eggs. This suggests the effect of size but not of OH.

Larger pupae have more OAc and a longer lifespan

Small pupae have more OAc but a shorter lifespan. This suggests the effect of size but not of OAc.

We analyzed the data in a multiple regression framework, where analysis of covariance and statistical interactions, respectively, account for and explain how different predictor variables interact. The reviewer is right that females with higher pupal mass have higher fecundity and are also the ones driving the correlation between rates of OH and fertility in Fig 4. However, this correlation exists AFTER accounting for the effect of pupal mass on fertility. Thus, there is both an effect from pupal mass and an effect from OH and, following condition-dependence expectations, this relationship is stronger in the high pupal mass females.

Minor comments

Abstract

Line 18 – “...continue the maintenance of genetic variation in a signal trait, despite selection from mate preference”. This sentence is misleading, selection for mate preference (sexual selection) actually maintains high variance in the trait (the lek paradox).

Changed “despite selection from mate preference” to “that are under natural or sexual selection”

Line 29 – in both groups, we found.... Longevity to be correlated with pheromone amount-

However, for early maters in negatively correlated, and for late maters I negatively correlated. Should be emphasized.

Since this an abstract with word limits, we feel that we shouldn't add more details to keep the message relatively simple.

Line 30 “This study is the first to report a correlation between fitness and sex pheromone composition in moths. This is a misleading sentence as “correlation between fitness and sex pheromone composition in moths” was already reported in Harari, Zahavi, Thiéry 2011 (#39 in the reference list).

We respectfully disagree, as Harari et al. 2011 only found a correlation between size and the amount of the major component, so that it is unclear whether total pheromone amount or composition was related to size. In addition, size was their measure, not fitness. Harrari et al. linked fitness to calling effort (through a third variable, proximity to congeners), but not to pheromone properties. There is also Jaffe et al 2007 who did investigate composition, but again in relation to size, not survival or reproductive success. We thus feel that the sentence in Ln 30 is justified and the relevant nuances are discussed in the Introduction (lines 62-72).

Line 43 “additionally, in many organisms sexual signals are used to localize potentially suitable conspecific mates”. I presume you mean here assortative mating, choosing among conspecific or natural selection choosing conspecifics from interspecifics. It is not clear from the sentence. Nevertheless, in both cases, a citation is missing. The following sentence is not clear either, what are these so-called species recognition signals, those you defined earlier (line 42) as under directional selection? Or those used for “suitable conspecific mates” which are also under sexual selection (although not directional). This paragraph needs clarification.

To clarify, we now added “often referred to as species recognition” to the end of the first sentence (see line 45).

Line 65 –change to - see review in 30).

We respectfully would like to keep this as is, because both papers cited are review papers and we do not see the need to emphasize this.

Line 76 there are not many papers in moths but quite a few in chemical communication in mammals (See review in Penn 2002 or Boulet et al. 2010) and in other insects (for example

Rantala et al. 2003, Moore et al. 1995, 1997; Worden et al. 2000; Chemnitz et al. 2015; and in moths - Jaffe et al. 2007; Harari et al. 2011). Please clarify

We now clarify by changing our statement to: “we lack empirical insight into the relationship between sexual signal variation and fitness for chemical mate attraction in insects generally”. To clarify to the reviewer: We cite most of the papers suggested by the reviewer in this paragraph. However, the great work by Moore et al on the cockroaches does not give details on the relationship between chemical signals and fitness. In the work on moths, none of the publications investigated a relationship between pheromone composition and fitness. This is why we conclude with this statement. However, we tried to do justice in this paragraph to the great work that has provided insight into the condition-dependence of pheromone signals in moths and other insects. A reference to Rantala et al. 2003 was missing, and we thank the reviewer for pointing this one out, which is now included.

Line 83 – signaling activity is mentioned here for the first time but it was not defined anywhere and not how was it measured. Please clarify.

We have now added “i.e. the time spent sending the pheromone signal” in parentheses to this sentence (Ln 84). In the methods we give further details (line 148 onwards).

Line 106. Emergence was checked mostly during the scotophase. Do adults not emerge in the photophase? Or was it due to convenience working hours? Please clarify.

Indeed, adults mostly emerge in the last hours of the scotophase

Line 138 – Please explain why samples containing <10 ng are not reliable

Because < 10ng across 11 pheromone components means that all peaks but the major are barely detectable from the chromatogram.

Line 153 – to measure fecundity females were mated. How? Was there a possibility to choose a mate or you assigned a male to a female? This may affect the results. As females mate more than once, young females that were assigned to low-quality males may delay oviposition, awaiting a better male.

What was the males' age? Was it similar in both early and late maters? Old males may have less fertilizing sperm. Please clarify.

Females were assigned a mate and males were always between 2-3 days old, as specified in line 157. There is always the option that females could adjust their mating, fertilization and oviposition decisions based on the perceived male quality, but this cannot easily be accounted for. We did always include the pupal mass of the male mate as a covariate in our analyses.

Line 226. Old females lay fewer eggs than young females. The phenomenon is known. Here the assumption (line 226) is that the reduced fertility is due to loss of time – missing 5 days of laying, i.e. all females young and old are capable of laying X eggs/day. That is, females of old age do not have less energy or fewer viable eggs when are old. Is there any physiological support

for this? Alternatively, a similar mean number of eggs per day of old and young females is only a coincidence, maybe due to the distribution of female size in the two groups? Please clarify

We did not make an assumption, but found that late maters laid fewer total amount of eggs, but not fewer eggs per day. Late maters did have lower per day fertility (and lower total fertility). Based on these findings, we present two, non- mutually exclusive explanations for why there may be fewer eggs or offspring: less time to lay eggs and less energy. We now also acknowledge that reproductive senescence may play a role (see lines 375).

Line 325 – we also observed examples of effects in the opposite direction for both fertility and lifespan. What does that mean? How often?

This means that both negative and positive correlations between signal stability and fitness exist. The referenced supplementary figure allows one to count how often it occurs. Both positive and negative correlations have meaning, as they could be interpreted as condition-dependence and trade-offs, respectively. This is explained in the introduction (lines 53-61)

Lines 382-390. The arguments in the paragraph are not clear. 1) High relative amount of Z11-16:OAc is correlated with higher fitness of females (Line 287). (2) Apparently, Z11-16:OAc is only produced when communication interference is likely to happen (line 388). (3) In lab colony, in the absence of *H. virescens* for many generations Z11-16:OAc not only exists but also associates with higher fitness (line 391). (4) A surprising conclusion – “selection likely favors females that produce lower relative amounts of acetate esters when there is no risk of heterospecific mate attraction”. So, is more of the Z11-16:OAc related to better fitness in the lab population (1) or not (4)? If pheromones are costly to produce, why Z11-16:OAc is still produced after so many generations in the lab in the absence of *H. virescens*? What is the relative ratio of this component in the pheromone blend? Please clarify.

The relative percent of Z11-16:OAc in the pheromone blend ranges from 5 - 30 %. Our data suggest that in our lab populations, higher rates of Z11-16:OAc are associated with higher fitness. If the observed correlation in our data is indicative of a direct or indirect cost, this could mean that that females may not invest in (much) Z11-16:OAc when there is no risk of heterospecific male attraction. Since this component has a dual function, i.e. not only inhibiting attraction of heterospecific males but also increasing attraction of conspecific males (see e.g. Groot et al.2006), we do not expect this compound to become absent in lab populations.

Line 414. Stability is correlated with higher fitness, but a negative correlation was also found. This is puzzling. What ratio of the females had a positive effect on fitness and what ratio of negative effect?

The correlations describe the patterns across all females, but these patterns differ between fitness measurements and mating delay treatment group

Line 415. “The evidence thus supports our hypothesis for fitness costs to sex pheromone stability”. Again, the cost may be assumed but there is no evidence for the such.

To avoid overstating the results, we now removed this sentence and added “In line with our expectations, ...” to the sentence right before.

Line 417. “it is possible that female benefits from maintaining a young-female-like blend”. This is an interesting argument and should be elaborated. Does this mean that females cheat regarding their age. Do they also cheat regarding their fecundity? Alternatively, males do not care about females' age but their mind their fecundity, thus females advertise their fecundity, not their age. But, did stability correlate also with size? This may strengthen the argument of pheromone as an honest signal, as only females with high fitness may keep signaling with their quality despite their aging. Please clarify.

The idea put forwards in this paragraph is that if the correlations we find are manifestations of costly stability of a costly signal, then pheromone characteristics could be a reliable signal/cue of their age, similar to what is found in crickets and mice. Although this does raise the question about honesty and about the information males may get from the signal, this is beyond the scope of this study. So this will have to await further data, especially on male response behavior.

431- the relationship between a moth sex pheromone signal and fitness is already studied in Harari et al. 2011. Please clarify.

As we explained above, Harari et al. 2011 showed that female size and amount of the major component of *Lobesia botrana* were correlated. They further showed that females signal more on the first night (but not on later nights and not in total across nights) when exposed to conspecifics and that female exposed to conspecifics also have reduced fitness. At best this would allow for the conclusion that (i) pheromone amount of condition-dependent (but not that it is correlated to fitness) and that (ii) signalling effort (but not amount/composition) due to the presence of potential competitors trades-off with fitness. Although the paper is exciting and a great value to the discussion of condition-dependence of pheromone signalling, we believe there is no evidence of a relationship between signal properties and fitness. To clearly separate our results from the evidence in previous research, we now adjusted the corresponding lines in the summary paragraph as follows (39= Harari et al. 2011, 40 = Jaffe et al 2007, 41 = Rantala et al 2003, 42 = Chemnitz et al. 2015): “Our results are in line with earlier findings for condition-dependence of pheromone signals in moths and beetles (40–42) and for signaling activity-fitness trade-offs in moths (39). Our results, go beyond these findings by revealing (i) a relationship between a moth sex pheromone signal and fitness and (ii) finding this relationship not only for the amount of pheromone produced, but also for the composition and for the extent to which females keep their signal amount and composition stable.”

Appendix B

Dear Editor,

Please find enclosed the revised manuscript RSOS-210180.R1 "Sex pheromone signal and stability covary with fitness". We again appreciate the time and effort invested by you and by the reviewers. We have incorporated the three changes suggested by the reviewers:

R1:

- 1) I would like to give only an additional request; please add citation(s) of moth pheromone databases, such as the pherobase (<https://www.pherobase.com>) or the List of Pheromones and Attractants (<https://lepipheromone.sakura.ne.jp>) for your description in L69-70.

We now cite Pherobase in Ln 70

R3:

- 2) Line 152-153: females were mated in 475-mL observation cups with virgin 2-3-day-old males". Change to each female was mated in a 475-mL observation cup with a randomly chosen, virgin 2-3-day-old male..." or similar

We have changed the text accordingly

- 3) Line 375 – "...it is possible that a female benefits from maintaining a young-female-like blend. This would be similar to... (72,73)." "This", here, refers to the possibility that old females benefit from maintaining a young-female-like blend. However, this is not what the two papers discuss. Rephrase.

We have changed the sentence to: "Several aspects of animal signals may change with age, and the signals of younger individuals have been found to be more attractive to the receivers, for example in cricket songs and mouse urinary protein pheromones (73,74)." We also moved the sentence before the sentence starting with "Since in moths..." to disconnect the cited papers from our statement about the possibility of benefitting from a young-female-like blend.

In case you have any questions, please do not hesitate to contact me.

On behalf of all co-authors and with kind regards,

Thomas Blankers
University of Amsterdam